# Screening, Identification and Application of Lactic Acid Bacteria for Degrading Mycotoxin Isolated from the Rumen of Yaks

**DOI:** 10.3390/microorganisms12112260

**Published:** 2024-11-07

**Authors:** Youli Yao, Jinting Luo, Peng Zhang, Yongben Wang, Boyu Lu, Guofang Wu, Jianbo Zhang, Xuan Luo, Lei Wang

**Affiliations:** 1Key Laboratory of Qinghai Plateau Livestock Nutrition and Feed Science, Qinghai University, Xining 810016, China; 15709833353@163.com (Y.Y.); zhangpeng_1260@126.com (P.Z.); 13613828741@163.com (Y.W.); 2071440326@139.com (B.L.); 2Plateau Livestock Genetic Resources Protection and Innovative Utilization Key Laboratory of Qinghai Province, Xining 810016, China; jingtingluo@126.com (J.L.); qhuwuguofang@126.com (G.W.); zhangjb9122@163.com (J.Z.); cauluoxuan@163.com (X.L.)

**Keywords:** mycotoxins, lactic acid bacteria, probiotics, fermented feed

## Abstract

Mycotoxin contamination is a major food safety issue worldwide, posing a serious threat to animal production performance and human health. Lactic acid bacteria are generally regarded as safe fermentation potential probiotics. They have the advantages of low toxicity, small pollution, strong specificity and high safety, and can reduce the contamination of microorganisms and mycotoxins. In this study, we compared the mycotoxin degradation capacity of 15 lactic acid bacteria strains from the rumen of Qinghai yak, through comprehensive analysis, we finally identified the strains as potential probiotics because they have a fast growth speed, strong acid production capacity (pH < 4.5), and they can grow normally in an environment with a pH of 3.5, bile salt concentration of 0.1%, and good self-agglutination and hydrophobicity. It was found that the fermentation group (*Pediococcus acidilactici* C2, *Pediococcus acidilactici* E28, *Pediococcus pentosaceus* A16 and *Enterococcus lactis* C16) could significantly reduce mycotoxin content, and both the nutritional and fermentation quality of the feed improved after 7 days of fermentation, meaning that the fermentation group can be used as a functional feed additive.

## 1. Introduction

Mycotoxins can contaminate a variety of food and feed products, most severely in tropical and subtropical regions [1,2]. Leading to huge economic losses. According to the FAO, financial damage amounts to 25% of the global harvest [3]. Once ingested, mycotoxins can cause adverse reactions in both humans and animals. Therefore, exposure to mycotoxins should be minimized. In addition, the presence of mycotoxins in feed can lead to poor animal performance and cause animal-derived food safety problems, resulting in allergic reactions and death in animals or humans [4]. Common mycotoxins include Aflatoxin B1 (AFB1), Zearalenone (ZEA), Deoxynivalenol (DON), T-2 toxin (T-2), Ochratoxins A (OTA), Fumonisin (FB1), Patulin (PAT), Citrinin (CTN) and Ergot alkaloids (EA) [5], which can induce clinical or subclinical effects, including decreased food intake, impaired intestinal integrity, vaccination failure, liver overload, and damage immune system [6]. According to a recent global survey, 90% of feed or raw materials contain at least one mycotoxin and 64% of feed contains multiple mycotoxins. AFB1, ZEA and DON were detected in 92.4% of samples. This study showed that co-contamination with multiple mycotoxins is common [7,8].

Aflatoxin includes subtypes like B1, B2, M1, M2, G1 and G2, all having strong carcinogenicity, mutagenicity and teratogenicity. AFB1 has the highest toxicity profile, which can damage the animals’ livers, inhibit their immune functions, and lead to cancerous deaths in animals and humans [9]. The effects of AFB1 on poultry include weight loss, poor feed efficiency, reduced egg production, egg weight loss, increased liver fat, altered organ weight, decreased serum protein levels, damaged liver, suppressed immunity, and reduced activity of several enzymes involved in the digestion of starch, proteins, lipids, and nucleic acids [10]. AFB1 causes more severe liver damage in pigs than in poultry and can cause slow growth and even the death of piglets. Reducing feed intake and milk production in dairy cattle and increasing the number of somatic cells can lead to death [11]. Peng Xi et al. [12] reported that AFB1 could weaken the functions of thymus and bursa in broilers without affecting growth and development. Studies have found that the toxic mechanism of AFB1 is related to the reactive oxygen species, leading to cell membrane and DNA being damaged [13].

DON poisoning in livestock leads to slower growth, refusal to feed, nausea, vomiting, diarrhea, and can even damage the hematopoietic system, which can ultimately result in death [14]. DON can impair the brain and central nervous system of livestock and poultry, leading to increases in serotonin synthesis in the brain, which is one of the main causes of anorexia and vomiting in animals. Compared with pigs, poultry is less sensitive to DON and has no significant effect on their growth performance; ruminants have a strong tolerance to DON and their sensitivity to DON can vary with sex and age [15,16].

ZEA can affect the animal feed conversion rate, growth, and development, leading to abnormal animal behavior and infertility, false pregnancy, and abortion at high concentrations. Once pregnant animals consume food contaminated with ZEA, ZEA can cause abortion, stillbirth, and abnormalities [17]. Pigs are the most sensitive to ZEA. Sows show uterine expansion, breast swelling, reduced ovulation and a prolonged estrous cycle [18]. In addition, children will be precocious after eating such animal products; therefore, ZEA is also called mold estrogen [19].

In recent years, some researchers have selected probiotics that can reduce toxins or produce beneficial metabolites, such as *Streptococcus* and *Enterococcus*, screened by Niderkorn et al. [20], which can degrade Fusarium in corn silage to reduce the accumulation of toxicity in vivo. In addition, *Lactobacillus casei* L30 has the ability to bind to 49.2% AFB1 [21]. Furthermore, *Lb. rhamnosus* 753, which is isolated from corn and rape, can effectively inhibit the accumulation of AFB1 under hot and humid conditions, screened by Ma [22]. *Lb. brucei* [23] from corn silage has a remarkable effect on significantly improving ketosis in dairy cows screened by Li, This finding provides a novel approach for probiotics to enhance animal health and reduce the cost of breeding in the livestock industry.

Furthermore, a mixed strain of bacteria composed of *Pseudomonas*, *Bacillus licheniformis*, *Achromobacter* and *Flavobacterium* isolated from soil achieved the complete degradation of ZEA without leaving any toxic substances [24]. Zhang et al. [25] found that the enzyme present in the supernatant of the *B. licheniformis* BL010 strain has the remarkable ability to effectively degrade AFB1, which is a significant finding in the field of mycotoxin degradation. Guo [26] found that when *B. licheniformis* YB9 cultured in liquid broth medium for 48 h, it exhibits the best degradation effect on DON. The degradation rate reached 82.67%. This indicates that a specific strain duration is crucial for achieving optimal degradation of DON by this bacterium. The degradation rates of AFB1 and OTA by *B. licheniformis* CM 21 isolated from fermented soybeans in Thailand are 74.0% and 92.5%, respectively [27]. Dague et al. [28] demonstrated that when the pH of the medium increased from 2.5 to 6.0, the ability of *Bacillus lichenifsis* to remove ZEA increased.

Sangsila [29] found that the adsorption capacity of Lactic acid bacteria (LAB) increases with an increase in the initial concentration. Gratz et al. [30] found that exposure of *Lacticaseibacillus rhamnosus* GG to AFB1 reduced the binding of the toxin to the intestinal mucus, resulting in a faster remove toxin. *Lb. casei*, which was screened by Hernandez-Mendoza et al. [31,32], mainly produced cell wall acids and combined with AFB1 to form a stable toxin complex. The results confirmed that the detoxification rate of living *Lb.* cells to mycotoxins is 50–80%, but the adsorption efficiency reached 86–100% after all the microorganisms were inactivated, indicating that LAB may use the adsorption mechanism to form a bacterial polymerization with mycotoxin, rather than removing mycotoxins through extracellular enzymes [33,34]. In addition, intestinal bacteria (*Lb. acidophilus*) and plant-related bacteria (*Lb. plantarum* and *Lb. brevis*) have been shown to have high degradation rates [35].

Although some progress has been made in the current research on the degradation or adsorption of mycotoxins by probiotics, there is still a need to find effective new technologies, new methods, and new strains. Based on this, this experiment screens lactic acid bacteria that can efficiently degrade mycotoxins and have potentially probiotic properties from 169 strains of lactic acid bacteria isolated from the rumen of Qinghai yaks in the early stage, laying a foundation for the application of functional potentially probiotics.

## 2. Materials and Methods

### 2.1. Sample Collection and Isolation

Ruminal contents were collected from naturally grazing Qinghai yaks at different growth stages in Guinan County, which is situated at an altitude of over 3100 m in Qinghai Province, China. Samples were collected in sterile centrifuge tubes in October 2019, and quickly transported to Xining in dry ice and stored in an ultra-low-temperature refrigerator at −80 °C. Ruminal contents samples (1 g) were added to 9 mL of sterilized water and then diluted to 10^−2^, 10^−3^, 10^−4^, 10^−5^, 10^−6^, 10^−7^ fold diluted and inoculated on MRS-Ca_2_CO_3_ agar, which was incubated at 37 °C for 48 h. The suspected colonies were picked and further purified with streak plate method. The above steps were repeated 4 to 5 times until a single colony was obtained. The purified single colonies were stored in an MRS slant culture medium at 4 °C until use. Subcultures were performed every 4 weeks for further analysis.

### 2.2. Biological Characteristics of LAB Strains

To evaluate the characteristics of the LAB strain, the growth and acid production curves, measurements and physiological and biochemical tests were performed. The bacterial solution to be tested was inoculated in MRS liquid medium according to 3% inoculum, 37 °C constant temperature, and the OD_600_ values and pH values were measured at 0 h, 2 h, 4 h, 6 h, 8 h, 12 h, 16 h, 24 h, 36 h and 48 h, and the results were recorded. Growth curves and acid production curves were drawn using the time of bacterial culture as the abscissa and the absorbance and pH values as the ordinate.

### 2.3. Screening for Highly Degradation Toxin LAB Strains

In order to screen LAB, which can effectively degrade mycotoxins and anti-nutritional factors, first, evaluate the degradation ability of 169 isolates for each of the four mycotoxins and select the top ten for each. Then, focus on the top 20 with a relatively good degradation ability for all four mycotoxins, and test their simultaneous degradation ability. Finally, select 15 selected isolates tested for the degradation rates of aflatoxin, zearalenone, glucosinolates and deoxynivalenone. Firstly, to prepare the fermentation fluid, 900 μL of fermentation broth was added into a 1.5 ml centrifuge tube, with 100 μL of 10 mg/L AFB1 standard solution, 10 mg/L DON standard solution, 10 mg/L of GLUS standard solution and 25 mg/L ZEN standard solution to avoid light incubation. Measure the toxin content at the following time points: 0 h, 2 h, 4 h, 8 h, 12 h, 1 d, 3 d, 7 d and 14 d. Take the culture time as the abscissa and the degradation rate as the ordinate to draw the degradation curve. Analyze the change in toxin degradation rate in different time periods and screen the strains with the highest degradation rate in a short time period. The samples’ content of AFB1, DON, GLUS and ZEN were measured with an ELISA kit to calculate the mycotoxin degradation rate.

### 2.4. Screening for Antimicrobial LAB Strains

The Oxford cup double-layer plate method is outlined by Cui et al. [36]. *Escherichia coli ATCC 30105*, *Salm. enterica ATCC 43971*, *Staphylococcus aureus ATCC 29213* were selected as pathogenic bacteria, which were activated and cultured in LB medium. They were cultured at 37 °C for 24 h and set aside. To evaluate the inhibitory activity, 200 µL of LAB cell-free supernatant was added to each Oxford cup and incubated at 37 °C for 24 h. The diameter of the inhibition zone was measured to compare the efficacy of the strains. 

### 2.5. Hemolysis Test and Antibiotic Resistant Test

All screened strains were streaked on Columbia agar plates and incubated for 24 h under anaerobic conditions. The hemolytic effects were recorded by the presence of a clear zone (β-hemolysis), a green zone (α-hemolysis) or the absence of a zone (γ-hemolysis) around the colony. Antibiotic susceptibility assays were performed according to the method described by Eliana L S D [37]. The candidate strains were evenly coated on MRS solid medium, and 8 drug-sensitive papers, including erythromycin (E), ofloxacin (OFX), streptomycin (S), penicillin (P), tetracycline (TE), gentamicin (GEN), polymyxin B (PB), ciprofloxacin (CIP), etc. were pressed with sterile tweezers to stick the surface of the medium. After each plate had 4 pieces evenly placed on it, the antibiotic name was marked 37 °C inverted to culture for 24 h, and the antibacterial coil diameter was measured by the vernier caliper.

### 2.6. Acid Tolerance and Bile Salt Tolerance

The acid tolerance test was performed following the protocol proposed by Guo et al. [38]. To assess the acid tolerance, LAB was inoculated with 1% inoculum in MRS liquid with different pH. Using HCl adjust the pH of the medium to pH (2.5, 3.0, 3.5, 4.0 and 6.0), cultivated at 37 °C for 24 h, the OD_600_ was measured and compared with the uninoculated media with the corresponding pH as the control. Similar to the method of acid tolerance testing, the bile salt tolerance test on the growth of the selected isolates was carried out by applying the method of Argyri et al. [39]. The activated strains were inoculated with 1% inoculum in MRS liquid media containing different concentrations of bile salts (0 g/L,1 g/L, 2 g/L and 3 g/L). After that, they were cultivated at 37 °C for 24 h, Subsequently, the OD_600_ was measured and compared with the uninoculated media with the corresponding concentration of bile salts serving as the control.

### 2.7. Auto Aggregation Test and Cell Surface Hydrophobicity

The auto aggregation test according to the method was described by Li [40]. The candidate strains were incubated for 24 h at 37 °C in the MRS broth medium and were centrifuged (8000× *g*) for 5 min at room temperature. The obtained cell pellets were resuspended, washed three times with PBS and the absorbance of the bacterial suspension was adjusted. We put 4 mL of the cell suspension adjusted absorbance into centrifuge tubes and incubated at 37 °C for 5 h. Auto aggregation was determined by measuring the absorbance OD_600_ at the beginning of the incubation (A_0_) and after 5 h (A_t_). At the same time, we took 3 mL of the cell suspension adjusted absorbance into centrifuge tubes, added 1 mL of xylene and precultured for room temperature for 10 min. We then quickly Vortex mixed for 2 min, let stand at room temperature for 15 min to layer, absorbed the lower aqueous phase, and measuree the OD_600_ value (At). The calculation of the self-agglutination rate and surface hydrophobic rate of *Lb.* cells: A% = (A_0_ − A_t_)/A_0_ × 100%.

### 2.8. Identification of the Selected Strains by 16S rRNA Gene Sequence Analysis

The DNA of the LAB was extracted according to the kit D3350-02 instructions of OMEGA (Omega Biotek, Norcross, GA, USA). After extracting the bacteria DNA, PCR was performed with the primers of 16S F (5′-AGA GTT TGA TCC TGG CTC AG-3′) and 16S R (5′-GGT TAC CTT GTT ACG ACT T-3′). PCR reaction system (25 μL) was composed of 1 μL of DNA template, 1 μL of 16S F, 1 μL of 16S R (10 μmol/L) each, 12.5 μL of 2 × Taq PCR Master mix, and 9.5 μL of ddH_2_O. PCR amplification program was set as follows: pre-deformation at 94 °C for 3 min, 40 cycles of denaturation at 94 °C for 40 s, annealing at 55 °C for 40 s, and extension at 72 °C for 30 s, followed by the extension at 72 °C for 10 min. The PCR products were sequenced by the Huada Biotech Company (Zhengzhou, China). The homologies between the gained sequences and those in GenBank were evaluated using BLAST (https://blast.ncbi.nlm.nih.gov (accessed on 11 July 2023)) analysis on the NCBI. Typical strain sequences with more than 97.00% similarity were selected, and the phylogenetic tree was constructed using MEGA software (https://www.megasoftware.net (accessed on 15 July 2023)).

### 2.9. Fermentation Feed Production

In our fermentation feed production, specific raw materials such as corn, wheat bran, extruded soybeans, sunflower meal, and corn gluten meals were used for fermentation, which was purchased from Hu Zhu County (Qinghai, China). The standby feed is sprayed with water and exposed to the air for a week to cause mildew and then stirred evenly.

The four microorganisms *Pediococcus acidilactici* C2, *Ped. pentosaceus* A16, *Enterococcus lactis* C16, and E28 were cultured for 24 h independently and mixed at a ratio of 1:1:1:1; the viable count of the mixed bacterial solution is greater than 2.35 × 10^9^ CFU/mL. Next, we diluted the mixed bacterial solution with water so that the mixed bacterial solution accounted for 5% of the diluted solution. Finally, we mixed 500 g of mildew feed with 500 mL of diluted solution (including 5.0% mixed bacteria), divided it into a 500 g fermentation bag, and vacuumed sealed and cultured at 37 °C. The test treatment design is as follows: (1) CK (control group); (2) Z14 (test group, C2, A16, C16, E28). Each group had three replicates. Samples were collected from each treatment group at 0, 1, 3, 5, 7, 10 and 15 days, A total of 42 samples were obtained. After sampling, the fermentation quality, nutritional quality, and microbiological changes were determined.

### 2.10. Determination of Mycotoxins in Fermented Feed

A total of 5 g of feed was dried and crushed and mixed with 10 mL of 70% methanol. The mixture was vigorously shaken for 3 min and then filtered with filter paper. The filtrate was collected, and the contents of AFB1, DON, ZEN and GLUS were determined by ELISA (Shanghai, China).

### 2.11. Analysis of Feed Fermentation Quality and Nutritional Quality

After sampling on days 0, 1, 3, 5, 7, 10 and 15, the samples were immediately stored in a sealed sample bag at 20 °C for further analysis. The pH of the samples was measured using a pH meter (Mettler Toledo Co., Ltd., Greifensee, Switzerland). The samples were dried in a 65 °C oven for 48 h to determine the dry matter (DM). The concentration of lactic acid, acetic acid, and propionate in the samples was determined by using a liquid chromatograph HPLC1200, and the butyrate concentration was determined by gas chromatograph GC6890. Crude protein (CP, GB5009.5-2016), neutral washing fiber (NDF, GBT20806-2006), the content of acid washing fiber (ADF, NYT1459-2007) was determined by the corresponding national standard method. Ammonium nitrogen (NH_3_-N), soluble sugar (SS), starch (ST), lipase (LPS), cellulase (CL), acid protease (ACP), α-amylase (α-AL), β-amylase (β-AL) were determined using a kit produced by Shanghai Enzyme-linked Biotechnology Co., Ltd. (New Taipei City, Taiwan)

### 2.12. Analysis of the Bacterial Communities

After the total DNA of the sample was extracted, primers were designed according to the conserved region, and sequencing joints were added to the end of the primers, PCR amplification was performed, and the products were purified, quantified and homogenized to form sequencing libraries. The constructed libraries were first inspected, and qualified libraries were sequenced by Illumina Nova-Seq 6000. The primer sequence of the V3 + V4 region is F: ACTCCTACGGGAGGCAGCA; R: GGACTACHVGGGTWTCTAAT. First, the Trimoraic v0.33 software was used to filter the sequenced Raw Reads. Then, cut adapt 1.9.1 software was used to identify and remove the primer sequences to obtain the Clean Reads that do not contain the primer sequences; the dada2 method in QIIME2 2020.6 was used to denoise and remove the chimera sequences to obtain the final effective data.

### 2.13. Statistical Analysis

All experiments were performed in triplicate. The data are expressed as mean ± SD. The mean and standard deviation (SD) were calculated, and other statistical analyses were carried out in Microsoft R Excel 2010 software package and SPSS 20.0.

In this experiment, SPSS 20.0 software was utilized for data processing and statistical analysis, and Origin 2017 was employed for drawing. A one-way ANOVA was applied to test and analyze the significance of differences (*p* < 0.05). Microbial community data were visualized through the bio-cloud Microbial Analysis Platform (https://international.biocloud.net accessed on 15 March 2024).

## 3. Results

### 3.1. Growth Curve and Acid Production Curve

The growth curves were plotted with the OD_600_ nm changes and pH values of the four LAB strains (Figure 1). The growth curves of the 15 strains exhibited sluggish, logarithmic, and stable stages. All strains were in the slow growth period between 0 and 4 h, with the exception of strain A16 which entered the logarithmic growth period after 6 hours of inoculation while the rest of the strains entered this stage after 4 hours. Strains B8 and D4 were in the stable growth period from 16 to 48 h, and the remaining strains were in this period from 24 to 48 h. At 48 h, the OD_600_ value ranged between 1.5 and 2. Among them, strains B8, D4 and D3 had the highest OD_600_ value and demonstrated strong growth and reproduction ability. In addition, the pH value of all the bacteria dropped to below 4.5 at 48 h.

### 3.2. Detoxication Curves

The degradation rates of all lactic acid bacteria on four toxins showed an upward trend over time (Figure 2). The degradation trends of 15 strains of lactic acid bacteria for AFB1 were almost the same. They rose rapidly after 12 hours. The degradation rates of strains B8, C1 and A16 for AFB1 were 81.08%, 80.12% and 78.96%, respectively, on the 14th day. Degradation trends of the 15 strains of lactic acid bacteria for DON were almost the same as well. They gradually stabilized after 12 hours. On the 14th day, the degradation rates of strains A7, C1 and B8 for DON were 69.82%, 68.89% and 68.38%, respectively. The degradation rate of ZEA in the 15 strains of lactic acid bacteria showed the same trend. The degradation rate of C16 was the highest at 14 d (73.56%). The degradation rate of strain E28 was the highest at 12 h (43.41%), and strain C2 was the highest at 24 h (53.84%). Strains C2, A16, C16, and E28 exhibited the highest degradation rates of AFB1, DON, ZEA and GLUS, respectively, at 33.53%, 60.18%, 51.45% and 43.41%, respectively.

### 3.3. Antimicrobial Activity

The fifteen tested strains were able to inhibit the growth of the indicator strains to different degrees (Table 1). Among them, strains D4, C1 and D17 showed strong antibacterial activity against *Staph. aureus* ATCC 29213. Except for strains B8, C24, C16 and E11, the other strains showed strong antibacterial activity against *E. coli* ATCC 30105. Strains B1, D4, C2, D3, C1 and E28 showed strong antibacterial activities against *Salm. enterica* ATCC 43971 Strains D4, C1, C2, D17 and E28 showed good inhibitory effects on the three indicator bacteria. These results indicated that our strains can synthesis substances with antibacterial activity, which reinforces their choice for possible use. *P. acidilactici* strains might have relatively higher antimicrobial activity against Gram-negative pathogenic bacteria compared to Gram-positive ones.

### 3.4. Hemolysis Test and Antibiotic Resistant Test

None of the 15 *Lb.* strains showed symptoms of hemolysis after 48 h of incubation (γ), and all 15 strains were insensitive to streptomycin and gentamicin (Table 2).

### 3.5. Acid Tolerance and Bile Salt Tolerance

To reach and colonize the intestine, LAB must have some tolerance to bile salts, which means high and low viability in the intestine [41]. Prolonged exposure of LAB to acidic conditions similar to those of the stomach was carried out by incubation at pH 2.0, 2.5, 3.0, 3.5, 4.0 and 6.0 for 24 h. Strains C2 and E28 exhibited strong acid and bile salt tolerance, respectively. Strains A16 and C16 were weakly resistant to acid and bile salts, respectively. Strains A16 and C16 can grow normally in acidic environments at pH 3.5 and pH 4.0. In a study on the acid tolerance of four strains of Enterococcus, it was found that two strains showed no survival at pH 2.0 and 2.5 during 2 h of incubation [42], whereas the four strains exhibited good viability at pH 3.0 (Figure 3).

### 3.6. Auto Aggregation Test and Cell Surface Hydrophobicity

In order to explore the adhesion abilities of different strains on epithelial cells, the hydrophobicity of their cell surfaces was evaluated in vitro. The results showed that among numerous strains, strain C24 stood out with its auto-aggregation rate, reaching as high as 47.13% (Figure 4A). Based on Sahil et al. [43], different strains of lactic acid bacteria genera, including Lactobacilli, exhibit differences in hydrophobicity rates on the cell surface. Strain B8 had the highest hydrophobicity rate of 79.46%, followed by strain E11, which had a surface hydrophobicity rate of 77.02% (Figure 4B).

### 3.7. Identification of the Selected Strains by 16S rRNA Gene Sequence Analysis

Based on the above results, strains C2, A16, C16 and E28 are dominant bacteria that can efficiently degrade mycotoxins in a short time and have good probiotic properties. Therefore, we performed 16S rRNA gene analyses for C2, A16, C16 and E28. BLAST was used to compare 16S rRNA gene sequences of the superior strains in GenBank. The results showed that the 16S rRNA gene sequences of the LAB strains were identified by more than 99%, and the phylogenetic tree was constructed using the neighbor-joining method (Figure 5). Strain A16 was identified as *P. pentosaceus*, strains C2 and E28 as *P. acidilactici* and strain C16 as *E. lactis*.

### 3.8. Determination of Mycotoxins in Fermented Feed

The changes in mycotoxins during feed fermentation are shown in Figure 6. With the exception of DON, the content of all three toxins decreased with the increase in fermentation days. AFB1, DON, ZEA and GLUS levels were lower in the Z14 group than in the control group. The AFB1 content of the Z14 group was significantly lower than that of the control group on the third day (*p* < 0.05) and was reduced by 50.73% (Figure 6A). The DON content decreased by 50.70% on the 15th day (Figure 6B). The ZEA content was significantly lower than that of the control group during fermentation (*p* < 0.05), and the lowest content was reached on the 7th day, which was reduced by 85.90% (Figure 6C). The GLUS content in the Z14 group was significantly lower than that in the control group on the third day (*p* < 0.05), and there was no significant difference at other times (Figure 6D).

### 3.9. Fermentation Quality and Chemical Composition

The fermentation quality and chemical composition of the feedstuffs were measured at seven time points: 0, 1, 3, 5, 7, 10 and 15 days. The LA, AA and BA contents in both groups increased overall, while PA decreased (Figure 7). Among them, the LA content of fermented feed in the treatment group (Z14) was significantly higher than that in the CK group at 10 d (*p* < 0.01), the AA content was significantly higher than that in the CK group before 7 d (*p* < 0.01), and significantly lower than that in the CK group after 7 d (*p* < 0.01).

The chemical compositions of the different treatment groups are presented in Table 3. With the number of fermentation days, the DM, CP, SS and ST content decreased, and the ADF, NDF and NH_3_-N increased. Moreover, the number of days of fermentation had a significant effect on the CP, NDF, ADF, NH_3_-N, SS, and ST content (*p* < 0.05), but there was no significant difference between the CP content in the Z14 and CK groups (*p* > 0.05). The DM content in the Z14 group was significantly lower than that in the CK group at the 10th d (*p* < 0.05), The ADF content was significantly lower in the 0 d, 3 d,5 d, and 7 d than in the CK group (*p* < 0.05), NDF content was significantly lower than the CK group at 15 d (*p* < 0.05), The NH_3_-N content was significantly higher in the 3 d, 5 d, 7 d, 10 d and 15 d than in the CK group (*p* < 0.05), SS content was significantly lower than the CK group at 1 d, 3 d, 5 d, 7 d, 10 d and 15 d (*p* < 0.05), ST content was significantly lower than the CK group at 3 d, 10 d and 15 d (*p* < 0.05), At other times, there were no significant differences between the two groups.

### 3.10. Microbial Diversity and Community Analysis During Fermentation

The non-multidimensional-dimensional scaling (NMDS) analysis diagram explains the changes in microbial communities by analyzing the beta diversity index (Figure 8). Greater similarity in community composition among samples would result in a more clustered appearance in the non-multidimensional scaling analysis diagram. In terms of bacterial composition, there was a certain distance between each treatment group; that is, there was a certain difference in bacterial composition between the control and treatment groups.

The bacterial community dynamics of the fermented feed at the phylum level are depicted in Figure 9A. In the original sample, *Firmicutes* (45.87%), *Proteobacteria* (16.28%), *Bacteroidetes* (5%) and *Actinobacteria* (5.89%) were the dominant phyla. In the fermentation process, *Firmicutes*, *Bacteroidetes* and *Actinobacteria* were dominant in all groups. However, specific community groups were affected by the fermentation treatment. Community composition changed during the fermentation process. The Z14 group showed a significantly higher relative abundance of *Firmicutes* than the CK group on the 7th and 10th day. Additionally, there was a relative abundance of *Proteobacteria* over time. At the end of fermentation, the Z14 group showed a significantly lower relative abundance of *Proteobacteria* than the CK group on the 15th day.

The bacterial community dynamics of the fermented feed at the genus level are shown in Figure 9B. In the original sample, “others” were dominant. Following others, *Ped.* (20.03%), *Bacillus* (9.35%) and *Lactococcus* (9.27%) were dominant at the genus level. After 7 days of fermentation, *Ped. Lb.*, *Bacillus*, and *Lc.* were higher in the Z14 group than in the control group. On the seventh day of fermentation, the relative abundance of *Limosilactobacillus* was 8.61% in the Z14 group, while the relative abundance of *Ls.* was close to zero in the CK group.

### 3.11. Microbial Species Differences Analysis and Correlation Analysis During Fermentation

The marker species with significant differences between groups were identified by lefse analysis (Figure 10), based on Chen [44]. The microbial community structures of the two groups gradually tended to be similar with increasing fermentation time. As for Z14 group, the species abundance of *Lactobacillaceae*, *Firmicutes*, *Ls.* and *Xanthomonadaceae* was more abundant in fermentation on the 1st, 7th, 10th and 15th days, respectively. Compared with the CK group, *Heynderickx* was the dominant genus for fermentation on the seventh day in Z14 group. However, probiotics became the dominant microbial community in the CK group after 15 days of fermentation.

The relationship between bacterial abundance and related indicators in the feed is shown in Figure 11. *Ped.* was positively correlated with ST, SS, GLUS, CP, ZEA, AFB1, pH and DM, but negatively correlated with NH_3_-N, ADF, β-AL, ACP, CL, α-AL, LPS, NDF and DON. *Lactococcus* was positively correlated with both PA and ZEA. PA was negatively correlated with *Lb.*, *Lentilactobacillus* and *Ls.* LA, AA and BA were positively correlated with *Acidovorax* and negatively correlated with *Lb*.

## 4. Discussion

Mycotoxins have hepatotoxic, mutagenic, nephrotoxic, immunosuppressive and carcinogenic effects, which can cause yield and economic losses worldwide. Animals that consume feed contaminated with mycotoxins can develop mycotoxin poisoning. Low concentrations of toxins can cause diseases such as liver and kidney lesions, en-terotoxic syndrome, adenogastric myogastritis and reproductive disorders, and myco-toxin residues in meat, eggs, and milk can easily lead to the risk of animal endogenous food safety. Therefore, effectively reducing mycotoxin pollution is important for improving animal production performance and human food safety [45]. However, Lb. can be used as a potential substitute for antibiotics owing to its high efficiency, strong specificity, and safety [46]. In addition, LAB can improve the intestinal microecological balance, promote growth and development, and improve immunity, antibacterial disease prevention, nutrition and health care functions [47]. Studies have shown that as a special ecological microenvironment, lactic acid bacteria screened and isolated from animal intestines are mostly probiotic and antibiotic sensitivity [48]. In addition, mixed fermented feed is often contaminated by fungi and mycotoxins, and biological control, which is one of the most effective fungal and mycotoxin treatment methods and has been widely reported as a research hotspot in recent years [49].

The results showed that 15 LAB strains showed varying degrees of degradation capacity against all four mycotoxins, which may be related to the special structure of the toxin; most toxic substances from organisms have corresponding degradation microorganisms in nature and may be secreted into the active substances leading to the biodegradation effect [50]. The results showed that *P. acidilactici* C2 degraded 70.95% of AFB1 at 72 h, *Enterococcus durans* B1 degraded 60.11% of DON, *Lb. rhamnosus* D3 degraded 60.64% of ZEA, and *E. lactis* C16 degraded 58.91% of GLUS. According to the report by Chlebicz et al. [36], the peptidoglycan, polysaccharides and teichoic acid in the cell wall of *Lactobacillus* mainly bind to AFB1 through hydrophobic interactions. Liżewska et al. [42] have reported that after 24 hours of fermentation by *Lb. paracasei* LOCK0920, *Lb. brevis* LOCK0944 and *Lb. plantarum* LOCK0945, the degradation rate of AFB1 can reach more than 60%. In addition, Zhang et al. [51] found that *L. helveticus* FAM22155 produces certain active proteins during solid-state fermentation, which jointly or individually degrade AFB1 to generate other substances. Zhao et al. [52] evaluated the adsorption capacity of 27 strains of *L. plantarum* for ZEN. The results showed that the adsorption capacities of the 27 strains for ZEN were different from each other, ranging from 1.72% to 47.80%. Moreover, the removal efficiency of ZEN was affected by factors such as bacterial density, initial toxin concentration, bacterial viability and incubation temperature. Haskard et al. [53] found that *Lb. rhamnosus* LC705 had a removal rate of 87.8% for AFB1 in PBS. *Lb. rhamnosus* GG had a removal rate of 84.1% for AFB1 in PBS. Chlebicz and Slizewska [54] showed that *Lactobacillus rhamnosus* 1088 had a removal rate of 79% for AFB1 in PBS.

Good bacterial measures such as acid production and rapid growth rate, whether in direct feeding or application in fermentation feed, enable lactic acid bacteria to quickly gain an advantage in the fermentation environment. With their strong acid production ability, they can form an acidic environment and produce a large amount of lactic acid, thereby preventing the reproduction of pathogens, as stated in references [55,56]. There exist differences and diversities among various lactic acid bacteria. These distinctions lead to different times for entering the logarithmic growth phase [57]. The acid production capacity is an important characteristic of lactic acid bacteria. Lactic acid bacteria produce organic acids such as lactic acid by fermenting carbohydrates, thus reducing the pH value of the environment. This not only helps to inhibit the growth of other harmful microorganisms but also affects their own metabolic and survival environments. For example, in the process of food fermentation, the acid produced by lactic acid bacteria can inhibit the growth of spoilage bacteria and pathogenic bacteria and extend the shelf life of food. At the same time, an acidic environment is also conducive to the growth and reproduction of lactic acid bacteria themselves because they can adapt to a lower pH value. Moreover, this is also the cause of the varying acid production capacities. In the study, on LAB strains isolated from Algerian traditional fermented products, Sawsen Hadef et al. [58] found that *Lactobacilli* showed stronger acidification ability than enterococci, and nine strains were *classified* as fast acidifying strains, and there were differences in aspects such as tolerance to different concentrations of NaCl. For the lactic acid bacteria that inhibit Aspergillus flavus screened from corn silage, the research by Gong et al. [59] found that, for example, the acid production rate of strain Q39 was the highest from 0 to 8 h, while the acid production rates of strains Q28 and Q44 reached the maximum from 6 to 8 h. There were differences in the time course of the acid production rates among different strains, which might be related to the differences in their metabolic pathways and enzyme activities.

Bile salt tolerance is one of the important indicators for screening probiotics. In the gastrointestinal environment, especially in the small intestine, bile salts can affect microorganisms. Strains with a bile salt tolerance can survive and function better in the intestine. In this experiment, LAB demonstrated excellent resistance to different bile salt concentrations. *Levilactobacillus brevis* C1 and *P. acidilactici* C2 had a strong tolerance. A study [60] discovered that *enterococci* strains isolated from camel milk showed viability at bile salt concentrations of 0.3%, 0.5% and 1.0%. For example, for the LAB strains screened from Bamei pig feces, in the test of simulating the gastrointestinal environment, its bile salt tolerance is one of the key factors for evaluating whether the strain can colonize in the intestine [61]. Some studies have shown that the bile salt tolerance of lactic acid bacteria may be related to the expression of certain proteins on the cell membrane. For example, the research by Hamon E et al. [62] found that in some Lb. strains, the bile salt tolerance is related to the expression of proteins such as GshR4, Cfa2, Bsh1, OpuA and AtpH. These proteins may help maintain the stability of the cell membrane, thus enabling the strain to survive in an environment where bile salts are present. Sawsen Hadef et al. [58] found that the BM10, B15 and C30 strains showed good tolerance to 0.5% bile salts. Among them, the C30 strain had the strongest tolerance, with a survival rate of 93.6 ± 1.6%, and there was no significant difference among these three strains (*p* > 0.5%). 

LAB is often considered a safe class of microorganisms; however, the improper selection of strains is a potential threat to livestock and poultry health. Animals may develop resistance to antibiotics, leading to greater treatment difficulty, which is also one of the reasons for screening probiotics for safety [63]. In this test, none of the 15 Lb. strains showed symptoms of hemolysis. Strains B1, C24, D4, D3 and E11 showed extensive drug sensitivity. The diverse inhibitory effects of different strains indicate that they have different inhibitory abilities. This may be related to the natural resistance genes carried by different strains and is more likely to be closely associated with the source of the strains. These results are in line with the studies of E. durans by Pieniz [64] and Reuben et al. [65]. 

Many microorganisms achieve the degradation of mycotoxins through the action of enzymes. For these four strains of lactic acid bacteria, there may be specific enzymes involved in the degradation process of mycotoxins. For example, the research by Madbouly Adel et al. [66] found that *Bacillus* can produce enzymes that degrade aflatoxin B1, and Tuncay [67] reported a novel flavoprotein in *Lysinibacillus sphaericus* that can enzymatically degrade aflatoxin B1. In addition to enzymatic reactions, strains may also interact with mycotoxins through components on its cell surface. For example, Niderkorn V et al. [68] studied the binding mechanism of lactic acid bacteria cell wall components with fumonisins. The four strains *P. acidilactici* C2, *P. pentosaceus* A16, *E. lactis* C16 and *P. acidilactici* E28 can degrade mycotoxins in a short time and possess good probiotic activity, which can be used in future studies.

The probiotic activity of lactic acid bacteria is affected by multiple factors. In terms of the strain’s own characteristics, its acid tolerance may originate from a special cell membrane structure or an acid-base regulation mechanism, enabling it to maintain the intracellular physiological balance and play a role in the acidic gastrointestinal environment. Similar to some studies, for example, Todorov [69] conducted a characteristic study on an acid-tolerant the *Lactobacillus plantarum* strain, which provides a reference for understanding the acid tolerance mechanism of lactic acid bacteria. The bile salt tolerance may be achieved by transporting bile salts out of the cell through cell membrane transporters or neutralizing the toxicity with substances inside the cell, thus allowing it to survive and colonize in the intestinal environment containing bile salts. Begley M [70] detailed the interaction between bacteria and bile, including the relevant mechanisms of bile salt tolerance, lactic acid bacteria strains with mycotox-in-degrading and probiotic activities have potentially wide applications in industry. In the feed industry, they can be used in the production of animal feed to reduce the m-cotoxin content in feed, decrease the impact of mycotoxins on animal health, and improve animal production performance [71]. Under practical conditions, although environmental factors may vary, these strains may still have a certain degree of effectiveness [72]. Due to their acid tolerance and bile salt tolerance, lactic acid bacteria can survive and colonize in complex environments such as the animal gastrointestinal tract, thereby exerting probiotic effects and degrading mycotoxins. In the industrial production environment, although environmental conditions such as temperature, humidity and nutrient components may fluctuate, through reasonable process control and strain screening optimization, these strains can be adapted to these conditions.

Feed supplementation using LAB is considered an effective way to improve feed quality. Studies have shown that adding LAB to feed can effectively increase the initial LAB load [73]. The decrease in pH during fermentation promotes microbial activity and fermentation [74]. A low pH can inhibit pathogenic bacterial growth, reduce protein loss, and extend shelf life [75,76]. In this study, pH decreased from 6.13 to 4.95 and remained within the normal pH range of its growth, meaning that multiplication and fermentation of LAB produced more lactic acid, which inhibited the growth of harmful bacteria. Furthermore, if lactic acid is insufficient, it provides a favorable opportunity for *Clostridium* growth, and its proliferation leads to the hydrolysis of sugars and proteins. Therefore, high-quality fermented feed depends on the rate of pH decrease [77]. Moreover, the ratio of LA to AA is an indicator indicating the quality of silage fermentation [78]. On the 10th d, the LA/AA ratio of the Z14 and CK groups was about 6:5, indicating that LAB fermented feed could effectively improve the feed fermentation quality, but the result on the 15th d was the opposite, which indicates that the fermentation time was not positively correlated with the feed fermentation quality, which may be caused by the increase in BA content.

Studies have reported that high DM content may affect the pH value, while low DM content leads to fermentation of harmful bacteria, and when DM content is less than 30%, it can promote the fermentation of Clostridium [79]. In addition, CP is an important indicator of mixed fermented feed; however, protein hydrolysis will inevitably occur during the fermentation process, in which proteins are hydrolyzed by plant proteases into peptides and free amino acids, and then further degraded into amides, amines, and ammonia through microbial activity, thus affecting the nutritional value of the feed [80]. In this study, there was no significant difference in DM (*p* > 0.05) among the different treatments at other time except for the 10th d, and the DM reached 41.65% on the 15th d, which was close to the ideal DM of good silage (40%). The number of fermentation days had a significant effect on the CP content. With the increase in fermentation time, CP content decreased from 17.29% to 15.37% within the period from day 0 to the 15th day, which may be due to the reduction in the total amount of CP caused by the release of carbon dioxide and water through respiration by beneficial bacteria that consume organic matter. These may be beneficial bacteria that consume organic matter that releases carbon dioxide and water through respiration. Studies have shown that feed with a CP content below 10% can lead to a decrease in rumen microbial activity, resulting in reduced gas production [81]. Moreover, an increase in NH3-N content indicates an increased activity of undesirable microorganisms; therefore, a high content of non-protein nitrogen in animal feed is undesirable. The results of this study showed that the NH3-N content of feed increased with fermentation time, which may be because the fermentation bag was not tightly sealed during the fermentation process, resulting in the decomposition of feed nutrients. NH3-N reached 4.33 mg/kg on the 15th days, but it was still within the allowable range, which was consistent with the change in NH3-N on the 7th–14th days when Fan et al. used silage to ferment wine lees [48].

In this study, the interaction between different treatment groups and different fermentation days significantly affected the ADF content. The ADF content of the Z14 group was consistently lower than that of the CK group at all the time points. The study found that the lower the NDF and ADF content, the better the feed quality, both of which were negatively correlated. Moreover, ST and SS contents decreased with time, which was due to the excessive LA content that consumed ST and SS. Studies have considered that this might be due to the LAB being active during fermentation, and the rapid acid production in the early stage of fermentation consumes energy, while ST and SS are utilized to maintain their own growth. This is consistent with the results of the present trial [82].

In this study, NMDS showed that the lactic acid bacteria fermentation group was significantly separated from the control group, indicating significant differences in the microflora structure between the two groups. This pattern was also reflected in the histogram of bacterial species abundance. During fermentation, *Firmicutes*, *Proteobacteria*, *Bacteroidetes*, and *Actinobacteria* were predominant at the phylum level. However, specific community groups were affected by the fermentation treatments and community composition varied with the fermentation process. Compared with the control group, the relative abundance of *Firmicutes* and *Bacteroidetes* in the Z14 group was always higher than that in the CK group at different times, which led to faster fermentation speed, while the abundance of *Proteobacteria* was lower than that in the CK group at different times, indicating that the fermented feed microecosystem became more stable, mature, and healthy with increasing time, which is consistent with previous studies [83,84]. *Firmicutes* and *Actinobacteria* have been found to degrade microbial cellulose in the animal body, promote the dissolution and digestion of fibers, and are the dominant bacteria involved in the process of cellulose decomposition. *Bacteroidetes* can improve the absorption and utilization of carbohydrates in animals, and they are the dominant bacteria involved in carbohydrate metabolism. However, the excessive relative abundance of Firmicutes can cause bacterial dysregulation; therefore, the abundance ratio of *Firmicutes* and *Bacteroidetes* plays a role in maintaining the stability of the microecoregion. However, if the relative abundance of *Firmicutes* is too high, it causes an imbalance in the flora. Therefore, the abundance ratio of *Firmicutes* and *Bacteroidetes* can maintain the stability of microflora.

At the genus level, *Pediococcus*, *Bacillus* and *Lc.* are considered the core bacteria in the process of hydrolysis and acid production [85], which can produce acetate, propionate or succinate as the fermentation end product [86]. On the 7th day of fermentation, the contents of *Ped.*, *Lb.*, *Bacillus*, *Lc.* and *Ls.* in group Z14 were higher than in the CK group. *Ls.* is a genus of Lb. involved in glucose metabolism with a high relative abundance. Therefore, dynamic changes in these genera are conducive to the production of VFA in the feed. Upon the completion of fermentation, within the intricate microbial ecosystem of the fermentation process, the relative abundance of the Z14 group at the genus level was significantly higher compared to the control group. This alteration in the microbial community structure has a profound impact on the overall fermentation dynamics. Among them, *Lb.* stands as one of the predominant genera. *Lb.* plays a crucial role in maintaining the stability and functionality of the microbial ecosystem during fermentation. It is highly beneficial to lactic acid fermentation by producing lactic acid through its metabolic activities, which in turn influences the pH and other environmental factors within the fermentation system. Moreover, its presence has a positive connection with the quality of the feed, as it contributes to the breakdown of complex substances in the feed and the formation of beneficial fermentation products that enhance the nutritional value and palatability of the feed.

Lefse result showed that the species abundance of *Lactobacillaceae*, *Firmicutes*, *Ls.* and *Xanthomonadaceae* were more abundant in feed microbial communities in fermentation on the 1st d, 7th d, 10th d and 15th d, respectively. Compared with the CK group, *Heymdrickxia* was the dominant bacteria of Z14 group fermentation on the 7th d, the study reported that *Bacillus* is non-toxic, non-residual, non-pathogenic bacteria, and it is drug resistant. Its antibacterial substances can promote the growth of livestock and poultry and it has a function in health care and the treatment of disease. In conclusion, the fermented feed with compound LAB had a significant effect on the microorganisms, and the Z14 group was better at 7 d compared to the CK group.

## 5. Conclusions

This study confirmed that inoculating the screened *P. acidilactici* C2, *P. acidilactici* E28, *P. pentosaceus* A16 and *E. lacti* C16 had positive effects on the fermentation process. Adding LAB could significantly improve the fermentation process by increasing the abundance of *Ped.* and the content of lactic acid, thus reducing the pH value. The Z14 groups exhibited significantly lower contents of mycotoxins than the control group, with a particularly significant reduction in DON contents. Furthermore, adding the LAB could reduce the relative abundance of Proteobacteria and *Acinetobacter*. In conclusion, the strains *P. acidilactici* C2, *P. acidilactici* E28, *P. pentosaceus* A16 and *E. lacti* C16 are appropriate choices for feed additives. This study provides a preliminary reference for practical production in the field of feed and food preservation. However, the mechanism of their reduction in fungi and mycotoxins needs to be further studied.

## 6. Patents

A kind of *E. lacti*, composition thereof and application thereof. Patent No.: CN202410035984.0. A strain of *P. acidilactici*, its composition and application. Patent No.: CN202410038368.0. A kind of *lactic acid bacteria* composition and its application Patent No.: CN202410038376.5. A strain of *P. acidilactici*, its composition and application. Patent No.: CN202410037522.2. A strain of *P. pentosaceus* containing its composition and application. Patent No.: CN 202410040487.X.

## Figures and Tables

**Figure 1 microorganisms-12-02260-f001:**
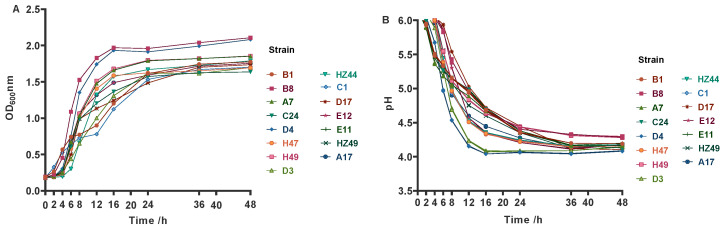
Growth (**A**) and acid production (**B**) curves of LAB strains isolated from the rumen of yaks.

**Figure 2 microorganisms-12-02260-f002:**
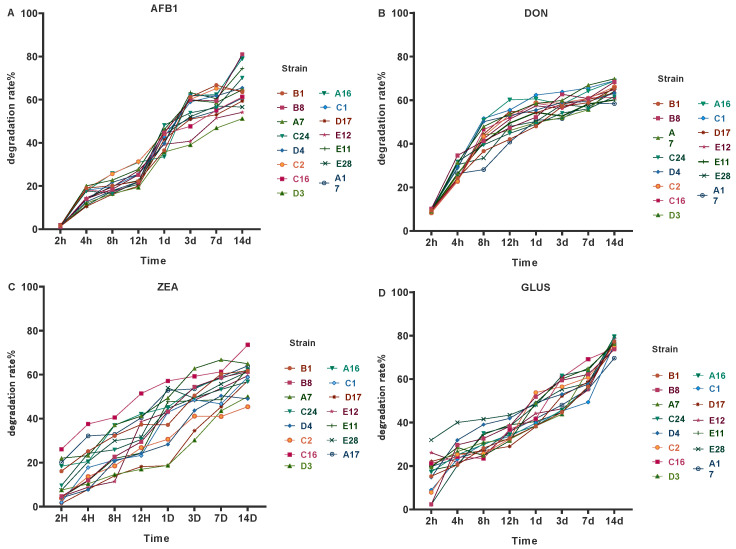
Detoxification curves of LAB strains isolated from the rumen of yaks. (**A**) represents the detoxification curve of the LAB strain(s) for AFB1 toxin; (**B**) represents the detoxification curve of the LAB strain(s) for DON toxin; (**C**) represents the detoxification curve of the LAB strain(s) for ZEA toxin; (**D**) represents the detoxification curve of the LAB strain(s) for GLUS.

**Figure 3 microorganisms-12-02260-f003:**
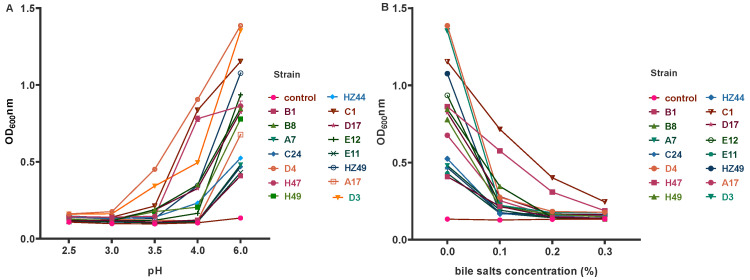
Growth of LAB strains isolated from rumen of yaks under different pHs (**A**) and bile salts (**B**).

**Figure 4 microorganisms-12-02260-f004:**
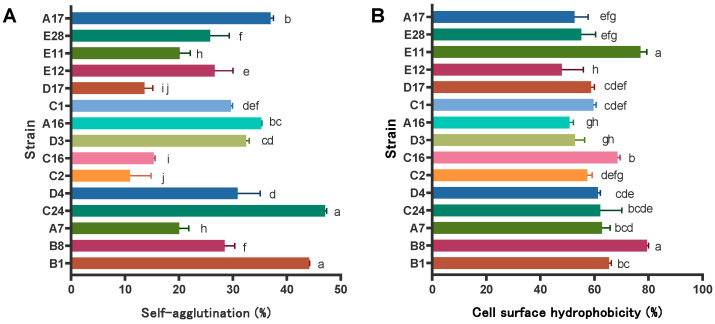
Self-agglutination (**A**) and cell surface hydrophobicity (**B**) of LAB strains isolated from the rumen of yaks. The same letter indicates no significant difference (*p* > 0.05), and different letters indicate significant difference (*p* < 0.05).

**Figure 5 microorganisms-12-02260-f005:**
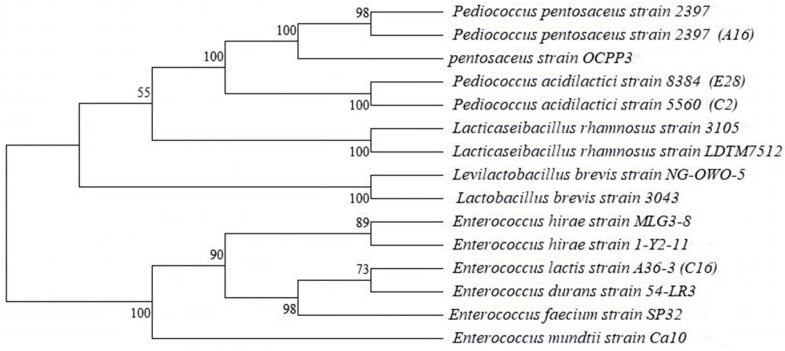
Phylogenetic tree of LAB strains isolated from rumen of yaks.

**Figure 6 microorganisms-12-02260-f006:**
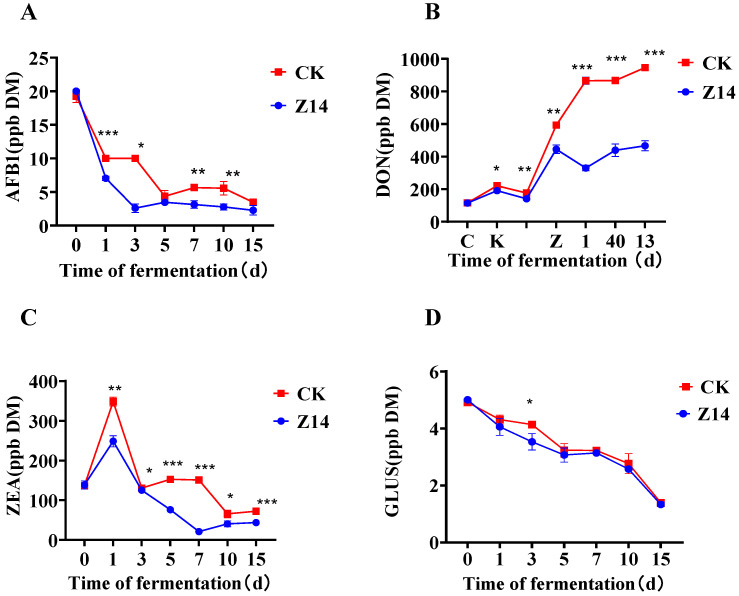
Changes in mycotoxin content during feed fermentation inoculated with a culture of four strains isolated from rumen of yaks. (**A**) indicates the change in AFB1 content; (**B**) indicates the change in DON content; (**C**) indicates the change in ZEA content; (**D**) indicates the change in GLUS content. Note: * *p* < 0.05, ** *p* < 0.01, *** *p* < 0.001; no * indicates that the difference was not significant. CK represents unfermented feed and serves as the control group without inoculation of yak rumen strains. Z14 represents fermented feed and is the experimental group inoculated with a culture of four strains isolated from rumen of yaks.

**Figure 7 microorganisms-12-02260-f007:**
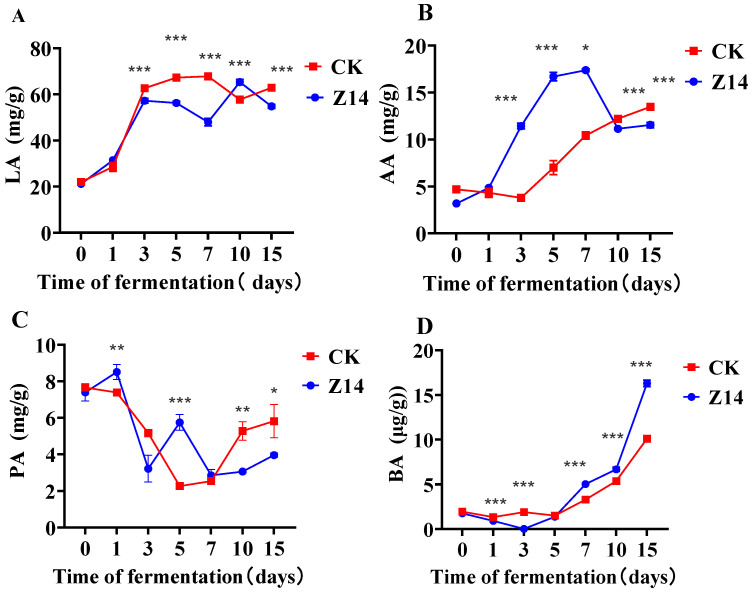
Changes in chemical parameters during feed fermentation inoculated with a culture of four strains isolated from rumen of yaks. (**A**) means Lactic Acid (LA); (**B**) means Propionic Acid (PA); (**C**) means Acetic Acid (AA); (**D**) means Butyric Acid (BA). Note: * *p* < 0.05, ** *p* < 0.01, *** *p* < 0.001; no * indicates that the difference was not significant.CK represents unfermented feed and serves as the control group without inoculation of yak rumen strains. Z14 represents fermented feed and is the experimental group inoculated with a culture of four strains isolated from rumen of yaks.

**Figure 8 microorganisms-12-02260-f008:**
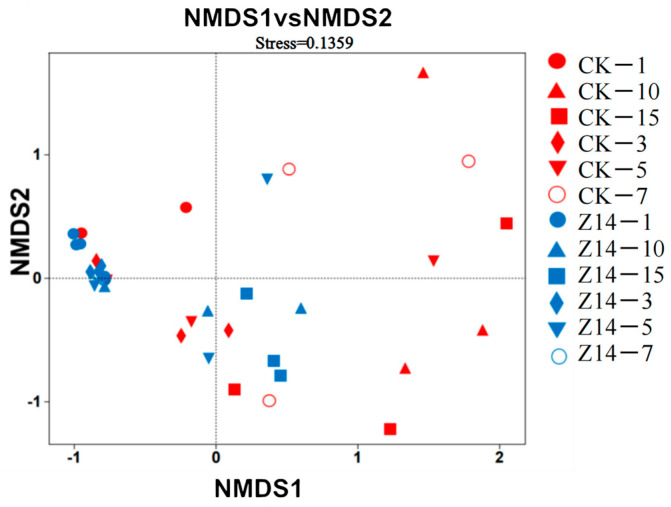
Non-multidimensional-dimensional scaling analysis during feed fermentation inoculated with a culture of four strains isolated from rumen of yaks. The blue ones represent the experimental group (fermentation group), while the red ones represent the control group (unfermented group). The same pattern in different colors indicates different treatments in the same period. CK represents unfermented feed and serves as the control group without inoculation of yak rumen strains. Z14 represents fermented feed and is the experimental group inoculated with a culture of four strains isolated from rumen of yaks.

**Figure 9 microorganisms-12-02260-f009:**
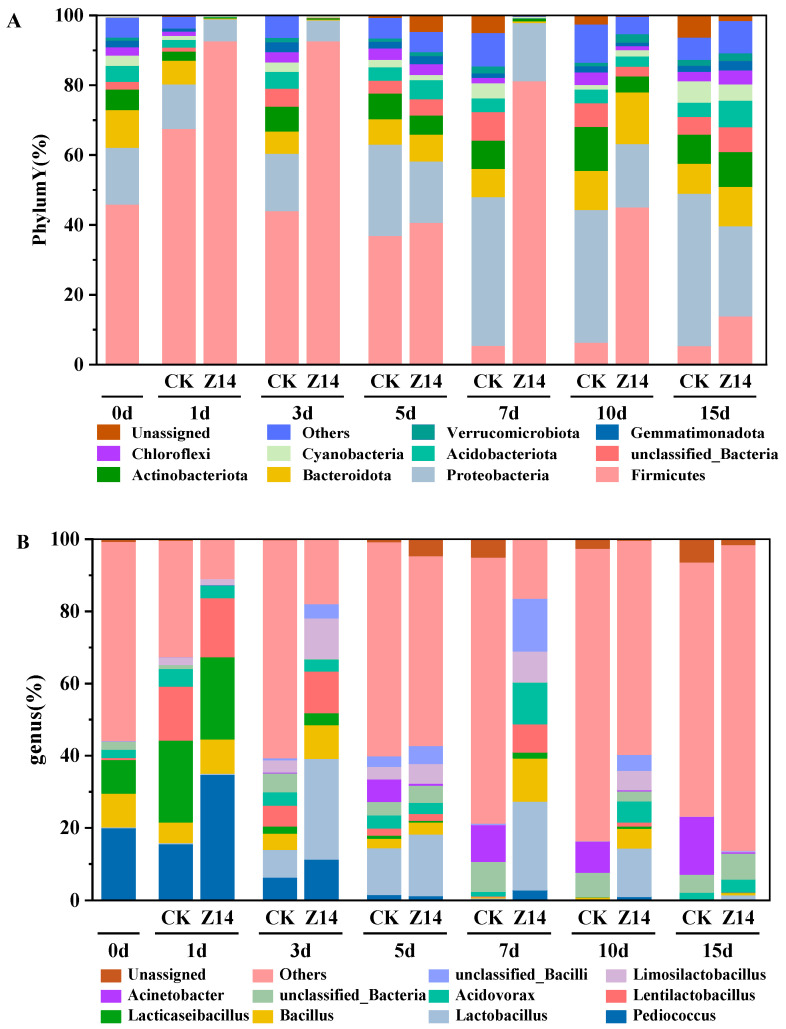
Bacterial community of feed during fermentation at phylum (**A**) and genus (**B**) levels. Note: CK represents unfermented feed and serves as the control group without inoculation of yak rumen strains. Z14 represents fermented feed and is the experimental group inoculated with a culture of 4 strains isolated from rumen of yaks.

**Figure 10 microorganisms-12-02260-f010:**
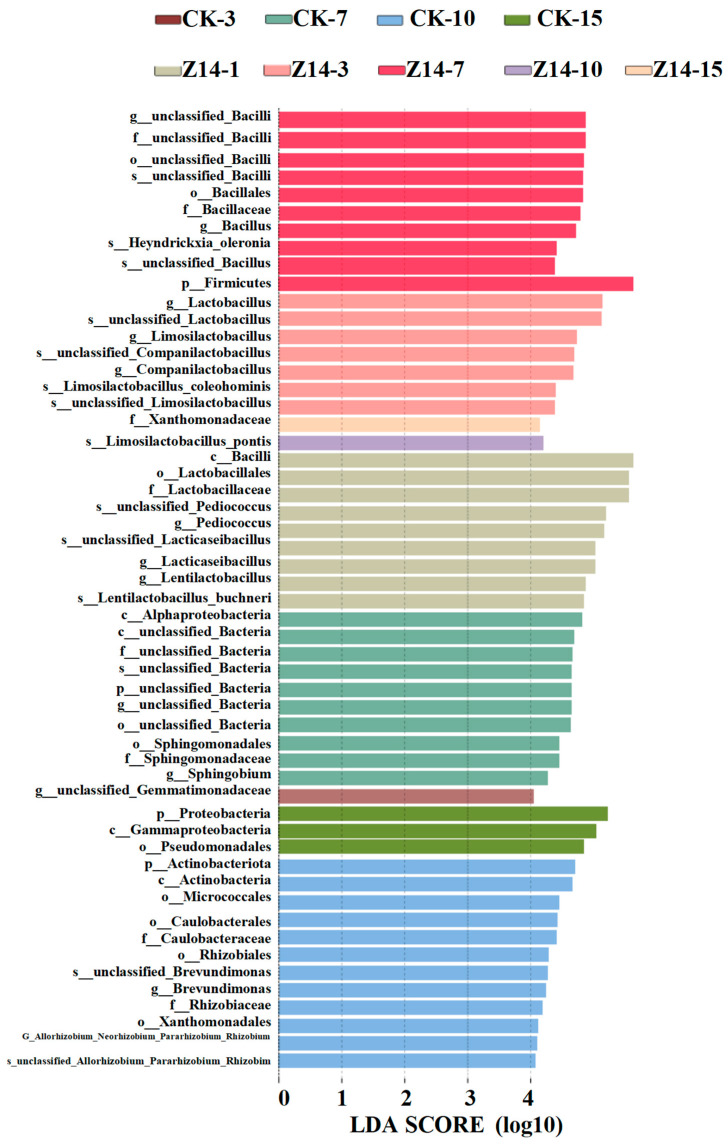
Lefse (Linear discriminant analysis Effect Size) Analysis.

**Figure 11 microorganisms-12-02260-f011:**
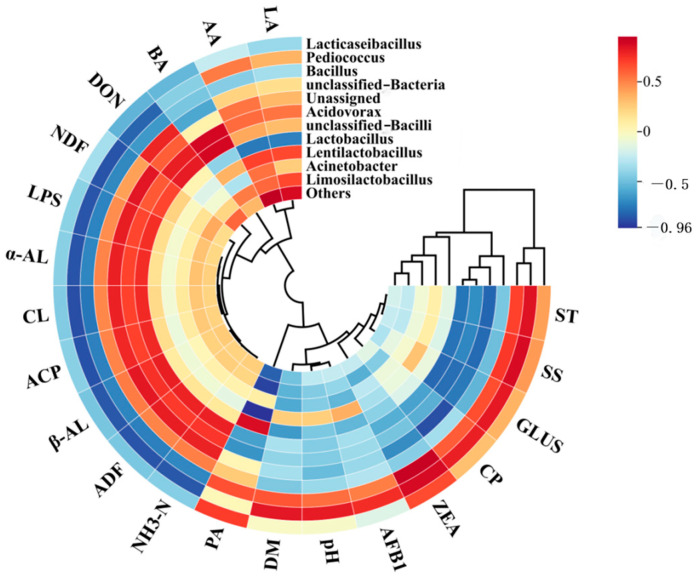
Microorganisms and feed quality correlation analysis.

**Table 1 microorganisms-12-02260-t001:** Antibacterial results of LAB strains isolated from rumen of yaks.

Strain	*Salm. enterica* ATCC 43971	*E. coli* ATCC 30105	*Staph. aureus* ATCC 29213
B1	++	++	+
B8	++	+	++
A7	+	++	++
C24	+	+	++
D4	++	++	+++
C2	++	++	++++
C16	+	+	++
D3	++	++	+
A16	+	++	++
C1	++	++	+++
D17	+	++	+++
E12	+	++	++
E11	+	+	++
E28	++	++	++++
A17	+	++	++

Note: “+” indicates that the diameter of the antibacterial zone is 9.0–12.99 mm; “++” indicates that the diameter of the antibacterial zone is 13.0–16.99 mm; “+++” indicates that the diameter of the antibacterial zone is 17.0–19.99 mm; “++++” indicates that the diameter of the antibacterial zone is greater than or equal to 20 mm.

**Table 2 microorganisms-12-02260-t002:** Hemolytic activity and antibiotic sensitivity of LAB strains isolated from rumen of yaks.

Strains	Hemolysis	Antibiotic Sensitivity							
		E	OFX	S	P	TE	GEN	PB	CIP
B1	γ	I	I	R	I	S	R	I	I
B8	γ	S	I	R	I	I	R	I	R
A7	γ	S	S	R	I	S	R	R	S
C24	γ	S	S	R	S	S	R	I	S
D4	γ	I	S	R	S	S	R	I	I
C2	γ	I	R	R	I	I	R	R	R
C16	γ	S	R	R	S	I	R	I	R
D3	γ	S	S	R	S	S	R	I	I
A16	γ	S	R	R	S	S	R	R	R
C1	γ	S	R	R	S	S	R	R	R
D17	γ	S	R	R	I	I	R	I	R
E12	γ	S	R	R	S	I	R	I	R
E11	γ	S	S	R	S	S	R	I	S
E28	γ	I	R	R	S	I	R	R	R
A17	γ	I	R	R	I	R	R	I	R

Note: R, resistant; I, intermediately susceptible; S, susceptible.

**Table 3 microorganisms-12-02260-t003:** Changes the nutritional quality during feed fermentation inoculated with a culture of four strains isolated from rumen of yaks.

Index	Groups	0 Days	1 Days	3 Days	5 Days	7 Days	10 Days	15 Days
DM (%)	CK	43.19 ± 1.54 ^a^	42.94 ± 2.00 ^a^	42.09 ± 0.82 ^a^	41.79 ± 1.25 ^a^	41.14 ± 1.26 ^a^	42.85 ± 0.31 ^a^	40.63 ± 0.79 ^a^
	Z14	43.83 ± 0.79 ^a^	42.48 ± 0.49 ^b^	41.65 ± 1.13 ^bc^	41.03 ± 0.65 ^c^	41.31 ± 0.13 ^bc^	41.81 ± 0.38 ^bc^*	41.65 ± 0.83 ^bc^
CP (%)	CK	17.33 ± 0.28 ^a^	16.74 ± 0.12 ^abc^	16.84 ± 0.50 ^ab^	16.17 ± 0.49 ^bcd^	16.02 ± 0.40 ^cd^	15.46 ± 0.11 ^de^	14.87 ± 0.66 ^e^
	Z14	17.29 ± 0.20 ^a^	16.60 ± 0.20 ^ab^	16.66 ± 0.61 ^ab^	16.12 ± 0.55 ^bc^	15.13 ± 0.42 ^d^	15.37 ± 0.48 ^cd^	14.87 ± 0.45 ^d^
ADF (%)	CK	19.26 ± 1.01 ^d^	20.40 ± 1.30 ^d^	21.15 ± 1.31 ^cd^	22.76 ± 1.07 ^bc^	22.94 ± 0.78 ^bc^	24.20 ± 1.47 ^ab^	25.08 ± 0.92 ^a^
	Z14	15.38 ± 0.71 ^f^	16.35 ± 1.11 ^ef^*	17.84 ± 1.31 ^de^*	19.12 ± 1.07 ^cd^*	20.30 ± 1.33 ^bc^*	21.86 ± 0.8 ^ab^	23.44 ± 1.60 ^a^
NDF (%)	CK	28.98 ± 1.16 ^e^	30.41 ± 1.67 ^de^	30.96 ± 0.67 ^cde^	31.72 ± 1.8 ^bcd^	33.64 ± 1.56 ^ab^	33.34 ± 1.18 ^abc^	34.46 ± 0.88 ^a^
	Z14	28.31 ± 0.72 ^d^	30.28 ± 0.87 ^c^	31.59 ± 1.57 ^bc^	32.77 ± 1.50 ^b^	35.20 ± 0.77 ^a^	35.43 ± 1.14 ^a^	37.11 ± 0.61 ^a^*
NH_3_-N (mg/g)	CK	2.09 ± 0.04 ^f^	2.34 ± 0.11 ^e^	2.55 ± 0.06 ^d^	2.82 ± 0.11 ^c^	3.05 ± 0.09 ^b^	3.20 ± 0.14 ^b^	3.46 ± 0.04 ^a^
	Z14	2.07 ± 0.01 ^g^	2.47 ± 0.02 ^f^	2.81 ± 0.03 ^e^*	3.17 ± 0.08 ^d^*	3.41 ± 0.03 ^c^*	3.87 ± 0.11 ^b^*	4.33 ± 0.10 ^a^*
SS (mg/g)	CK	39.91 ± 1.00 ^g^	37.56 ± 0.70 ^f^	34.00 ± 0.98 ^e^	31.75 ± 0.42 ^d^	28.71 ± 0.86 ^c^	26.52 ± 1.57 ^b^	23.99 ± 1.56 ^a^
	Z14	39.46 ± 0.31 ^g^	34.99 ± 0.89 ^f^*	30.67 ± 0.23 ^e^*	25.93 ± 0.39 ^d^*	22.29 ± 1.43 ^c^*	17.67 ± 1.70 ^b^*	13.17 ± 2.10 ^a^*
ST (mg/g)	CK	417.08 ± 10.19 ^a^	389.12 ± 10.22 ^b^	364.71 ± 3.07 ^c^	331.37 ± 7.75 ^d^	300.31 ± 10.71 ^e^	274.58 ± 3.63 ^f^	244.84 ± 3.44 ^g^
	Z14	409.52 ± 1.39 ^a^	374.38 ± 15.51 ^b^	347.48 ± 8.04 ^c^*	321.98 ± 11.39 ^d^	286.64 ± 3.9 ^e^	250.66 ± 2.21 ^f^*	219.68 ± 8.83 ^g^*

Note: The same column (*) or row (a–g) indicates a significant difference in the mean value (*p* < 0.05). DM, dry matter; CP: Crude Protein; ADF: Acid Detergent Fiber; NDF: Neutral Detergent Fiber; NH_3_-N: Ammonia Nitrogen; SS: Soluble Sugars, ST: Starch. CK represents unfermented feed and serves as the control group without inoculation of yak rumen strains. Z14 represents fermented feed and is the experimental group inoculated with a culture of four strains isolated from rumen of yaks.

## Data Availability

Upon reasonable request, and subject to review, the authors will provide the data that support the findings of this study.

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
