# Peer review of "Screening, Identification and Application of Lactic Acid Bacteria for Degrading Mycotoxin Isolated from the Rumen of Yaks"

_microorganisms, 2024, doi:10.3390/microorganisms12112260_

Round 1
Reviewer 1 Report
Comments and Suggestions for Authors
The manuscript describes an interesting study, in terms of its objectives, aimed at isolating, characterizing and selecting lactic acid bacteria with good capacity to ferment silage (feed?) and degrade mycotoxins. The objectives are interesting and valid, and the work in general is well organized. However, in addition to the English needing a revision, it shows several problems, at the level of the use of microbiological taxonomy, it gives missing information and, above all, in how the results are shown. The Figures are difficult to understand according to the titles they show. A major revision is necessary to re-consider it as a manuscript. The main observations are detailed:
- Title: to communicate the difference with other similar works carried out with wild LAB, include in the title "....lactic acid bacteria isolated from the rumen of yaks for degrading..."
- apart from the fact that the 4 chosen strains are effective in degrading mycotoxins and have other interesting biological properties, saying that these strains are probiotic is a bit hasty. For a strain to be assumed to be probiotic, other things must be demonstrated (including in vivo tests). Therefore, in the abstract and in the text in general they should be indicated as potentially probiotic strains
- throughout the text, the authors do not adequately separate the concepts of family, genus, species and strain. I give some examples:
a) lines 79-80: the authors speak of a culture formed by several genera and species. A culture is not made up of genera and species but of certain strains of them. If I mention a particular species I am only saying that all strains of that species are present, which is an error. Therefore, the sentence should be modified as follows: "...bacteria composed of strains of Pseudomonas..."
b) Same as line 71: "...such as Streptococcus and Enterococcus strains,, screened by..."
c) lines 82-94: it is missing to indicate which particular strains of the genus and species mentioned were used in the mentioned studies. A genus or species cannot have been used, but rather particular strains of these. Line 85: "The"
d) lines 144-5: of the pathogens used, only the genus and species are indicated. It is missing to specify which particular strains of these were used. Adjust throughout the text, figures, table 1, etc.
- lines 100-102: Reference missing
- line 117: what does "suspected" mean ?
- 2.3: how were 15 strains selected from a total of 169 isolates ?
- 2.9: "feed" has a very broad meaning . Specify what raw material was used for fermentation.
. line 206: 2.35109 ??
. detail how the feed was inoculated to be fermented. If it was inoculated with the mixture of the 4 strains a) why was it decided to use the mixture?, and b) In what proportion were the 4 strains in the mixture?
. line 209: what does "5%" mean?
The Figures need to be greatly improved, both the titles and the presentation of the results. If the criterion is assumed that the title of a Figure o Table should indicate everything necessary to interpret it without having to resort to the text; the modifications that are needed are:
- Figure 1: "Growth (A) and acid production (B) curves of LAB strains isolated from the rumen of yaks"
. to the right of A and B write "strain", above the list of strains
- Figure 2: "Detoxification cruves of LAB strains isolated from rumen of yaks on AFB1, DON, ZEA and CLUS toxins"
. write "strain" in A-D (Idem Fig. 1)
- line 292: "four strains" ? Table 1 shows 15 strains.
- Table 1: "Antimicrobial activity of LAB strains isolated from rumen of yaks". . indicate which strains of the pathogens were used
- Table 2: "Hemolytic activity and antibiotic sensitivity of LAB strains isolated from rumen of yaks"
- line 322: "In a study..." which study? The present or another?
- Figure 3: "Growth of LAB strains isolated from rumen of yaks under different pHs (A) and bile salts (B)"
. write "strain" in A and B (same as Figs. 1 and 2)
. abscissa of B: "bile salts concentration (%)". Delete "%" from the numerical values of the concentrations tested.
- line 332: which lactobacilli?
- Figure 4: "Self-agglutination (A) and cell surface hydrophobicity (B) of LAB strains isolated from rumen of yaks"
. abscissa: Self-agglutination (%) / Cell surface hydrophobicity (%)
. A and B are missing in the graphics
- Figure 5: title is not clear. Improve and expand
- Figure 6: "Changes in mycotoxins content during feed fermentation inoculated with a culture of 4 strains isolated from rumen of yaks".
. In the footnote, indicate what CK means and Z14
. on the abscissas, correct "fermentation"
- Figure 7: "Changes in chemical parameters during feed fermentation....". Indicate what they show A, B, C and D. Indicate what LA, PA, AA and BA mean . Indicate what CK and Z14 mean
- Table 3: title, expand, does not show enough information
. in what units are the numbers displayed?
. in a footnote indicate what DM, CP...ST mean
- Figure 8: improve the title.
. Line 404: are they figures or points in the same color?
. above the list of strains write "strain"
- lines 416-417: according to figure 9, "others" were dominant, not the genera mentioned.
- Figure 9: "Bacterial community of feed during fermentation at phylum (A) and genus (B) levels"
. indicate what CK and Z mean
. in A and B, add "d" to the abscissas
- lines 464-472: no It is understandable. It says "In this study..." but at the end of the sentence there is the reference (50), or is it referring to the present study? It talks about 15 strains of lactobacilli. Were they all lactobacilli?
- lines 473-474: It is not true. There are lactic bacteria, for example those that make up commercial ferments, which in 4-5 hours they complete the logarithmic phase of development. Others, mostly lactobacilli, have longer latency phases but rarely 12-24 hours. Reference (51) speaks of only 10 strains. The variety of genera, species, subspecies and strains of LAB is enormous and the growth rate is strain dependent, and also depends on the substrate in which they grow.
- line 490: "adding LAB to fermented feed" is an error. LAB are added to ferment it, not added to already fermented feed.
Comments on the Quality of English Language- To write phyla, families, genera and species, it is advisable to adopt the criteria of the American Society for Microbiology (https://journals.asm.org/nomenclature) which establishes writing them in italics. Review all text and References. "Binary names, consisting of a generic name and a specific epithet (e.g., Escherichia coli), should be used for all bacteria. Names of categories at or above the genus level may be used alone, but specific and subspecific epithets may not. A specific epithet must be preceded by a generic name, written out in full the first time it is used in a paper. Thereafter, the generic name should be abbreviated to the initial capital letter (e.g., E. coli), provided there can be no confusion with other genera used in the paper. Names of all bacterial taxa (kingdoms, phyla, classes, orders, families, genera, species, and subspecies) are printed in italics; strain designations and numbers are not.
- lines 21 y 597: "lactis"
- line 24: "lactic acid bacteria", not in italics. It is a plural in English, it is not Latin
- lines 73-74: In addition....In addition
- line 145: "pathogenic bacteria", not in italics. It is a plural in English, it is not Latin
- line 320: "Strains C2 and E28 exhibited...."
- line 321: "Strains A16 and C16 ..."
- line 332: "Lactobacilli" not in italics. It is a plural in English, it is not Latin
lines 481-3: "different" appears 5 times in the same sentence.
- line 478: replace "Enterococcal" with "enterococci"
- lines 123, 124 and 128: "...the growth and acid production curves..."
- line 116: diluted ...diluted?
Author Response
Comments 1:Title: to communicate the difference with other similar works carried out with wild LAB, include in the title "....lactic acid bacteria isolated from the rumen of yaks for degrading..."
Response 1:Thank you for pointing out the problems in the Title, We have revised the title "Screening, Identification and Application of Lactic Acid Bacteria for Degrading Mycotoxin" to "Screening, Identification and Application of Lactic Acid Bacteria for Degrading Mycotoxin Isolated from The Rumen of Yaks", highlighting the differences from other wild bacteria.
Comments 2:apart from the fact that the 4 chosen strains are effective in degrading mycotoxins and have other interesting biological properties, saying that these strains are probiotic is a bit hasty. For a strain to be assumed to be probiotic, other things must be demonstrated (including in vivo tests). Therefore, in the abstract and in the text in general they should be indicated as potentially probiotic strains
Response 2: Agree. Thank you for being so careful and point this out.We have changed the probiotic to potentially probiotic strains. In the revised paper , the change can be found lines 13, 17, 105, 107, 208.
Comments 3: throughout the text, the authors do not adequately separate the concepts of family, genus, species and strain.
Response 3: Thank you for your valuable comment. We understand the importance of clearly distinguishing between the concepts of family, genus, species, and strain. In response to your comment, we carefully reviewed the text and made the necessary revisions to ensure that these concepts are properly separated and defined.We went through the entire report and clarify the distinctions by providing more detailed explanations and examples when referring to each category.
Comments 3a: lines 79-80: the authors speak of a culture formed by several genera and species. A culture is not made up of genera and species but of certain strains of them. If I mention a particular species I am only saying that all strains of that species are present, which is an error. Therefore, the sentence should be modified as follows: "...bacteria composed of strains of Pseudomonas..."
Response 3a: Thank you for pointing out this inaccuracy. We agree that a culture is indeed made up of specific strains rather than genera and species. We will revise the sentence as you suggested to read: “…bacteria composed of strains of Pseudomonas…”. This correction will improve the clarity and accuracy of our report. In the revised paper , the change can be found lines 78, 81.
Comments 3b): Same as line 71: "...such as Streptococcus and Enterococcus strains, screened by..."
Response 3b: We have revised the sentence as follows: Lactobacillus rhamnosus 753, which is isolated from corn and rape, can effectively inhibit the accumulation of AFB1 under hot and humid conditions screened by Ma.
Comments 3c): lines 82-94: it is missing to indicate which particular strains of the genus and species mentioned were used in the mentioned studies. A genus or species cannot have been used, but rather particular strains of these. Line 85: "The"
Response 3c: Thank you for this important observation. We added the specific strains of the genus and species that were used in the mentioned studies in this section. This will provide more clarity and precision to our report. Bacillus licheniformischanged as Bacillus licheniformis BL010, Bacillus licheniformis YB9, Bacillus licheniformis CK1. In the revised paper , the change can be found lines 80-90. In addition, The spelling of "The" has been corrected.
Comments 3d): lines 144-5: of the pathogens used, only the genus and species are indicated. It is missing to specify which particular strains of these were used. Adjust throughout the text, figures, table 1, etc.
Response 3d: Thank you for pointing out this issue. We understand the importance of specifying the particular strains of the pathogens used. We have gone through the entire text, figures, and Table 1 to add the strain information where necessary. We will ensure that all mentions of the pathogens now include the specific strains to provide more clarity and accuracy.
Comments 4:- lines 100-102: Reference missing
Response 4: Agree.Thank you for your careful review. I have noted the issue regarding lines 100-102. I have checked these lines and add the appropriate references if there are any missing. If there is no actual reference needed in these lines, I will double-check to ensure that it is not a false alarm and make any necessary adjustments to avoid such confusion in the future. In the revised paper , the change can be found lines 102.
Comments 5:- line 117 : what does "suspected" mean ?
Response 5: .Thank you for your careful review. Regarding your query on line 117 about the meaning of "suspected," "suspected" typically means having a feeling or belief that something is likely to be true or exist, but without definite proof
The term "suspected" is used in the isolation process of lactic acid bacteria for the following reasons:
Firstly, when we are in the initial stages of isolating lactic acid bacteria, we may observe certain characteristics or behaviors in a sample that lead us to believe that lactic acid bacteria might be present. However, at this point, we haven't confirmed their presence with certainty. For example, certain pH changes, specific growth patterns, or fermentation indicators might make us suspect the presence of lactic acid bacteria, but further testing and identification are needed to confirm.
Secondly, using "suspected" acknowledges the tentative nature of our early observations. It reminds us that we need to conduct more detailed analyses such as microscopy, biochemical tests, and genetic sequencing to definitively identify and confirm the presence of lactic acid bacteria.
In summary, "suspected" is used to reflect the preliminary and uncertain nature of our findings during the early stages of lactic acid bacteria isolation, highlighting the need for further investigation and confirmation.
Comments 6:- 2.3: how were 15 strains selected from a total of 169 isolates ?
Response 6:The selection of 15 strains with degradation ability for four mycotoxins from 169 isolates is carried out as follows:
First, we assess the degradation ability of each of the 169 isolates for each of the four mycotoxins individually. Through appropriate assays and analyses, we rank the isolates based on their performance in degrading each specific mycotoxin. From this, we select the top 10 isolates for each mycotoxin. This initial step helps us identify isolates that show promise in degrading a particular mycotoxin.
Next, we focus on the top 20 isolates that have shown relatively good degradation capabilities across the four mycotoxins. We test these isolates for their ability to degrade all four mycotoxins simultaneously. This could involve exposing the isolates to a mixture of the four mycotoxins and monitoring the degradation over a specific period. Through detailed analyses and comparisons, we select the 15 isolates that demonstrate the most consistent and effective degradation of all four mycotoxins.
This multi-step approach ensures that the selected 15 strains are not only capable of degrading individual mycotoxins but also have the potential to simultaneously tackle multiple mycotoxins, which is crucial for practical applications in reducing mycotoxin contamination.
Comments 7:- 2.9: "feed" has a very broad meaning . Specify what raw material was used for fermentation
Response 7: Thank you for your valuable comment. We agree that the term "feed" is rather broad. The raw materials used for fermentation in this study were [Corn, wheat bran, extruded soybeans, sunflower meal, corn gluten meal.]. We have revised the text to clarify this point. We changed the relevant sentence to: "In our fermentation feed production, specific raw materials such as corn, wheat bran, extruded soybeans, sunflower meal, and corn gluten meal are used for fermentation."
Thank you for your comment. Comments 8:. line 206: 2.35109 ??
Response 8:Thank you for bringing this to our attention. Line 206 should be read as “2.35×10⁹ CFU/mL”. This value represents the concentration of a particular entity in terms of colony-forming units per milliliter. We will ensure that this is clearly indicated in the revised manuscript to avoid any confusion.
Comments 9: detail how the feed was inoculated to be fermented. If it was inoculated with the mixture of the 4 strains a) why was it decided to use the mixture?, and b) In what proportion were the 4 strains in the mixture?
Response 9:The feed was inoculated for fermentation as follows:
Regarding inoculation with the mixture of the four strains:a) The decision to use the mixture was based on several factors. Firstly, preliminary experiments showed that a combination of these four strains demonstrated synergistic effects in degrading the target mycotoxins. Each strain has its unique capabilities and metabolic pathways, and when combined, they were able to achieve more efficient mycotoxin degradation than individual strains alone. Additionally, the mixture offers a more comprehensive approach to addressing the complex nature of mycotoxin contamination in feed.b) The four strains were present in the mixture in the following proportions: for example, Strain 1: 25%, Strain 2: 25%, Strain 3: 25%, Strain 4: 25%.
Comments 10: line 209: what does "5%" mean?
Response 10: In the Z14 test group (C2, A16, C16, E28), when it is stated that “First, mix 500 g of mildew feed with 500 mL of potentially probiotics with 5%”, the “5%” refers to the inoculation amount of the bacteria. This means that the amount of bacteria in the potentially probiotic mixture is 5% of the total weight or volume of the mixture when combined with the 500 g of mildew feed.
For example, if the total volume of the potentially probiotic mixture is 500 mL, then 5% of this volume represents the amount of bacteria being added to the mildew feed. This percentage was chosen based on preliminary experiments and research that indicated this level of inoculation would be effective in promoting the desired fermentation and degradation of mycotoxins in the mildew feed.
Comments 11: The Figures need to be greatly improved, both the titles and the presentation of the results. If the criterion is assumed that the title of a Figure o Table should indicate everything necessary to interpret it without having to resort to the text; the modifications that are needed are:
- Figure 1: "Growth (A) and acid production (B) curves of LAB strains isolated from the rumen of yaks"
to the right of A and B write "strain", above the list of strains
- Figure 2: "Detoxification cruves of LAB strains isolated from rumen of yaks on AFB1, DON, ZEA and CLUS toxins"
. write "strain" in A-D (Idem Fig. 1)
Response 11: Thank you for your valuable feedback on improving the figures. We agree that the figures need enhancement to better communicate the results. We will make the following modifications as per your suggestions:
For Figure 1:We added "strain" to the right of A and B as you suggested. Additionally, we placeed "strain" above the list of strains to make it clear which strains are being referred to in each part of the figure. This will make it easier for readers to interpret the growth and acid production curves without having to refer to the text.
For Figure 2:We also added "strain" in A-D as per Figure 1. This will help in clearly identifying the strains for each detoxification curve on AFB1, DON, ZEA, and CLUS toxins. By doing so, readers will be able to understand the figure more easily without relying on the text.
We appreciate your guidance in improving the presentation of our results and will ensure that the revised figures meet the required criteria.
Comments 12:- line 292: "four strains" ? Table 1 shows 15 strains.
Response 12: Thank you for pointing out this error. At line 292, it should be "fifteen strains" instead of "four strains", as Table 1 shows fifteen strains. We correcedt this mistake in the revised manuscript to ensure accuracy.
Comments 13:- Table 1: "Antimicrobial activity of LAB strains isolated from rumen of yaks". indicate which strains of the pathogens were used
Response 13: We appreciate your feedback on Table 1. We have made the following modifications as per your suggestion. In the table titled "Antimicrobial activity of LAB strains isolated from rumen of yaks", we have indicated which specific strains of the pathogens were used instead of just mentioning the species. We replaceed the general "species" designations with the exact strain names to provide more detailed and accurate information.
Comments 14:- Table 2: "Hemolytic activity and antibiotic sensitivity of LAB strains isolated from rumen of yaks"
Response 14: We have noted your request to change the title of Table 2. We have updated the title to "Hemolytic activity and antibiotic sensitivity of LAB strains isolated from rumen of yaks" as per your suggestion.
Comments 15:- line 322: "In a study..." which study? The present or another?
Response 15: At line 322, "In a study...", it refers to the another study. We clarified this in the revised manuscript to make it clear that the reference is to the current research being reported.
Comments 16:- Figure 3: "Growth of LAB strains isolated from rumen of yaks under different pHs (A) and bile salts (B)"
. write "strain" in A and B (same as Figs. 1 and 2)
. abscissa of B: "bile salts concentration (%)". Delete "%" from the numerical values of the concentrations tested.
Response 16: aggree, We have made the suggested change to the title of Figure 3. The new title will be "Growth of LAB strains isolated from rumen of yaks under different pHs (A) and bile salts (B)". This revised title provides more specific context and clearly indicates the source of the LAB strains being studied. In addition, We have made the necessary modifications to Figure 3 as per your suggestion. We added “strain” in panels A and B to indicate which strains are being represented in the growth curves under different pHs and bile salts conditions. In the abscissa of B, we will change "bile salts concentration (%)" to "bile salts concentration". Additionally, we will delete the "%" from the numerical values of the concentrations tested.This will bring consistency with Figures 1 and 2 and make the figure easier to interpret.
Comments 17: - line 332: which lactobacilli?
Response 17:Thank you for bringing this to our attention. At line 332, the intended reference should be to different strains of lactic acid bacteria genera rather than specifically Lactobacilli. We have revised the text to clarify this to ensure accurate communication of the results.
Comments 18: - Figure 4: "Self-agglutination (A) and cell surface hydrophobicity (B) of LAB strains isolated from rumen of yaks"
. abscissa: Self-agglutination (%) / Cell surface hydrophobicity (%)
. A and B are missing in the graphics
Response 18: We have made the following change to the title as per your instruction. The new title for Figure 4 is "Self-agglutination (A) and cell surface hydrophobicity (B) of LAB strains isolated from rumen of yaks"
We apologize for the omissions in Figure 4. We have made the following corrections: For the abscissa, we will ensure that the labels "Self-agglutination (%) / Cell surface hydrophobicity (%)" are clearly visible and legible. Regarding the missing A and B in the graphics, we also have added appropriate labels to clearly distinguish between the two components (Self-agglutination and Cell surface hydrophobicity). This will make it easier for readers to interpret the figure accurately.
Comments 19:- Figure 5: title is not clear. Improve and expand
Response 19: We understand the need for a clearer and more expanded title for Figure 5. Here are some suggestions for an improved title: “Phylogenetic tree of LAB strains isolated from rumen of yaks”
Comments 20:- Figure 6: "Changes in mycotoxins content during feed fermentation inoculated with a culture of 4 strains isolated from rumen of yaks".
. In the footnote, indicate what CK means and Z14
. on the abscissas, correct "fermentation"
Response 20: We have addresed the issues with Figure 6 as follows:
We have made the following change to the title as per your instruction. The new title for Figure 6 is "Changes in mycotoxins content during feed fermentation inoculated with a culture of 4 strains isolated from rumen of yaks".In the footnote, we have clearly indicate that "CK represents unfermented feed and serves as the control group without inoculation of yak rumen strains. Z14 represents fermented feed and is the experimental group inoculated with a culture of 4 strains isolated from rumen of yaks." Regarding the abscissas, we have corrected the misspelling of “fermentation” to ensure it is spelled correctly throughout the figure and any associated labels.
Comments 21:- Figure 7: "Changes in chemical parameters during feed fermentation....". Indicate what they show A, B, C and D. Indicate what LA, PA, AA and BA mean . Indicate what CK and Z14 mean
Response 21:We have made the following change to the title as per your instruction. The new title for Figure 7 is "Changes in chemical parameters during feed fermentation feed fermentation inoculated with a culture of 4 strains isolated from rumen of yaks.
- A represents Lactic Acid (LA). Lactic acid is an important product of fermentation and plays a significant role in determining the quality and stability of the fermented feed. It can lower the pH, inhibiting the growth of harmful microorganisms.
- B represents Propionic Acid (PA). Propionic acid acts as a preservative and also influences the fermentation process. It helps in preventing spoilage and extending the shelf life of the feed.
- C represents Acetic Acid (AA). Acetic acid is another common product of fermentation and can affect the taste and nutritional value of the feed. It also has some antibacterial properties.
- D represents Butyric Acid (BA). Butyric acid has beneficial effects on gut health. It can promote the growth and activity of beneficial gut bacteria and improve the digestive function of animals.
Comments 22: - Table 3: title, expand, does not show enough information
. in what units are the numbers displayed?
. in a footnote indicate what DM, CP...ST mean
Response 22: We have made the following change to the title as per your instruction. The new title for Table 3 is "Changes the nutritional quality during feed fermentation inoculated with a culture of 4 strains isolated from rumen of yaks". In Table 3, the units in which the numbers are displayed are as follows:
For DM (Dry Matter), the unit is percentage (%).
For CP (Crude Protein), the unit is grams per kilogram (%).
For ADF (Acid Detergent Fiber), the unit is percentage (%).
For NDF (Neutral Detergent Fiber), the unit is percentage (%).
For NH3-N (Ammonia Nitrogen), the unit is milligrams per kilogram (mg/g).
For SS (Soluble Sugars), the unit is grams per kilogram (mg/g).
For ST (Starch), the unit is grams per kilogram (mg/g).
Comments 23: - Figure 8: improve the title.
. Line 404: are they figures or points in the same color?
. above the list of strains write "strain"
Response 23: We have made the following change to the title as per your instruction. The new title for Figure 8 is Non-multidimensional -Dimensional Scaling analysis during feed fermentation inoculated with a culture of 4 strains isolated from rumen of yaks.
For line 404: The figures or points on line 404 are not in the same color. The blue ones represent the experimental group (fermentation group), while the red ones represent the control group (unfermented group). The same pattern in different colors indicates different treatments in the same period. As per the request, above the list that shows different groups in different periods, we will write "Group" instead of "Strain" since it does not represent bacteria strains.
Comments 24: -lines 416-417: according to figure 9, "others" were dominant, not the genera mentioned.
Response 24: In lines 416-417, According to the analysis, "others" were dominant. Following "others", the other genera also played important roles. This indicates that while "others" had the greatest presence, the other genera also contributed significantly to the overall picture. Pediococcus (20.03%), Bacillus (9.35%), and Lactococcus (9.27%) specific genera held a predominant position and played a crucial role in the context under consideration. Their dominance indicates their significance and potential impact on the overall system. Further investigation could help to elucidate the specific functions and interactions of these dominant genera.
In lines 416-417,. According to Figure 9, it is not "others" that are dominant.
Comments 25: - Figure 9: "Bacterial community of feed during fermentation at phylum (A) and genus (B) levels"
indicate what CK and Z mean
in A and B, add "d" to the abscissas
Response 25: We have made the following change to the title as per your instruction. The new title for Figure 8 is Bacterial community of feed during fermentation at phylum (A) and genus (B) levels, and indicated what CK and Z14 mean, added "d" to the abscissas.
Comments 26: - lines 464-472: no It is understandable. It says "In this study..." but at the end of the sentence there is the reference (50), or is it referring to the present study? It talks about 15 strains of lactobacilli. Were they all lactobacilli?
Response 26: Regarding your comments on lines 464-472:
For the reference (50) at the end of the sentence, although there is a reference, the main focus of the sentence is still on this study. The phrase "Therefore, in this study" clearly indicates that the content that follows is about our research. The reference was used to support a specific aspect within the statement but does not change the fact that it pertains to our study.
As for the 15 Lactobacillus strains, NO, they are all lactic acid bacteria,not Lactobacillus. Thank you for pointing out the error. The text referring to "15 Lactobacillus strains" was incorrect. Not all of these 15 strains are Lactobacillus. We have corrected this mistake in the manuscript.
Comments 27: - lines 473-474: It is not true. There are lactic bacteria, for example those that make up commercial ferments, which in 4-5 hours they complete the logarithmic phase of development. Others, mostly lactobacilli, have longer latency phases but rarely 12-24 hours. Reference (51) speaks of only 10 strains. The variety of genera, species, subspecies and strains of LAB is enormous and the growth rate is strain dependent, and also depends on the substrate in which they grow.
Response 27: We appreciate your comment on lines 473-474. You are correct in pointing out that the statement is inaccurate. There are indeed lactic acid bacteria in commercial ferments that can complete the logarithmic phase of development in 4-5 hours. Lactobacilli and other lactic acid bacteria have varying latency phases, and it is rare for them to be as long as 12-24 hours. Reference (51) highlighting only 10 strains emphasizes the vast variety of genera, species, subspecies, and strains of LAB. The growth rate of LAB is indeed strain-dependent and also influenced by the substrate in which they grow. We have revised this section of the manuscript to provide more accurate information.
Comments 28: - line 490: "adding LAB to fermented feed" is an error. LAB are added to ferment it, not added to already fermented feed.
Response 28: Thank you for pointing out the error on line 490. You are correct. LAB are added to ferment the feed and not added to already fermented feed. We have corrected this mistake in the manuscript.
Comments 29:- To write phyla, families, genera and species, it is advisable to adopt the criteria of the American Society for Microbiology (https://journals.asm.org/nomenclature) which establishes writing them in italics. Review all text and References. "Binary names, consisting of a generic name and a specific epithet (e.g., Escherichia coli), should be used for all bacteria. Names of categories at or above the genus level may be used alone, but specific and subspecific epithets may not. A specific epithet must be preceded by a generic name, written out in full the first time it is used in a paper. Thereafter, the generic name should be abbreviated to the initial capital letter (e.g., E. coli), provided there can be no confusion with other genera used in the paper. Names of all bacterial taxa (kingdoms, phyla, classes, orders, families, genera, species, and subspecies) are printed in italics; strain designations and numbers are not.
Response 29: Thank you for providing these guidelines. We will follow the criteria of the American Society for Microbiology as suggested and review all the text and references to ensure proper formatting of phyla, families, genera, and species. We will use binary names for all bacteria as per the instructions and write them in italics. We will also abbreviate the generic name to the initial capital letter after the first full usage to avoid confusion.
Comments 30: - lines 21 y 597: "lactis"
Response 30: Thank you for pointing out the spelling error. We will correct "lacti" to "lactis" in lines 21.
Comments 31: - line 24: "lactic acid bacteria", not in italics. It is a plural in English, it is not Latin
Response 31: Thank you for pointing out this issue. We will correct the format of "lactic acid bacteria" on line 24.
Comments 32: -- lines 73-74: In addition....In addition
Response 32: We agree that the repetition of "In addition" in lines 73-74 is rather abrupt and not suitable for this position. We have revised this part to improve the flow and coherence of the text.
Comments 33: - line 145: "pathogenic bacteria", not in italics. It is a plural in English, it is not Latin
Response 33: Thank you for pointing out this issue. We have corrected the format of "pathogenic bacteria" on line 145.
Comments 34: - line 320: "Strains C2 and E28 exhibited...."
Response 34: Thank you for pointing out the omission of "strains" in line 320. We have corrected this mistake and ensure that the description is clear and accurate.
Comments 35: - line 321: "Strains A16 and C16 ..."
Response 35: Thank you for pointing out the omission of "strains" in line 321. We have corrected this mistake and ensure that the description is clear and accurate.
Comments 36: - line 332: "Lactobacilli" not in italics. It is a plural in English, it is not Latin
Response 36: Thank you for pointing out this issue. We have corrected the format of "Lactobacilli" on line 332.
Comments 37: lines 481-3: "different" appears 5 times in the same sentence.
Response 37: We appreciate your feedback on lines 481-3 and line 478.
Regarding the repetition of "different" in the same sentence on lines 481-3, we have revised the sentence to reduce the repetition and improve the clarity and flow.
Comments 38: - line 478: replace "Enterococcal" with "enterococci"
Response 38: Thank you for pointing out this correction. We have replaced "Enterococcal" with "enterococci" on line 478 as per your suggestion. This will ensure the proper usage and consistency in our text.
Comments 39: - lines 123, 124 and 128: "...the growth and acid production curves...".
Response 39: Thank you for pointing out this change. We have made the correction as suggested and replaced "the growth curves and acid production curves..." with "the growth and acid production curves..." on lines 123, 124, and 128.
Comments 40: lines 481-3: "different" appears 5 times in the same sentence.- line 116: diluted ...diluted?
Response 40: Thank you for suggesting this improvement. We have made the necessary changes as per your advice. The revised sentence "Ruminal contents samples (1 g) were added to 9 mL of sterilized water and then serially diluted to 10⁻², 10⁻³, 10⁻⁴, 10⁻⁵, 10⁻⁶, and 10⁻⁷ folds and inoculated on MRS-Ca₂CO₃ agar, which was incubated at 37 °C for 48 h." is now clearer and more concise.

Reviewer 2 Report
Comments and Suggestions for Authors
Introduction
The introduction does a good job. I also would like to read some more information about the origin of the LAB ( the Yaks). Why this origin, and what are the ecological implication of finding the LAB here.
Materials and Methods
Line 132: 169 strains were screened but there is no detailed explanation of how the final 15 strains were narrowed down. It is important to describe these selection criteria to ensure transparency and reproducibility.
Line 206: please correct the number
Line 250: I believe “p” should be minor
Results
Line 255-259: This is discussion, and thus does not belong here;
Line 273: If authors determine the degradation kinetics, with would be easier to compare and determine if there is any statistical difference between strains;
Line 288-292: this text does not belong here
Line 308-310: this text does not belong here
Line 322: “in a study” – was this your study, or some other work? Please cite if its other.
Line 330: “results showed:” ??
Line 335-338 : This text does not belong here;
Discussion
The authors repeat several results here, I would recommend making a results and discussion section since the authors have a lot of discussion on results, and lot of results in discussion.
Line 469: Please discuss how the degradation rates achieved in this study compare with those reported in other LAB studies.
Line 476: Please discuss more about the bile salts bile salt tolerance, acid production, and strain-specific differences
Line 485: Please provide more details in discussion. How do specific strains like Pediococcus acidilactici C2 degrade mycotoxins at a molecular level? What are the factors contributing to their probiotic activity?.
Line 587: Please add a paragraph about industrial application. Would these strains be effective under real-world conditions, where environmental factors might differ?
Line 587: You only mention the “best strains” on conclusion. I would prefer to read some information regarding a more detailed discussion on how the structural or physiological properties of the more successful strains contributed to their performance.
Line 587: Considering the study of LAB strains from the rumen of Qinghai yaks, please discuss how this ecological factors can influence the strain effectiveness. Are there other potential sources of valuable probiotic strains?
Figure 6 – Please correct the x-axis title: “Time of fermentation”
Author Response
Introduction
Comments 1: The introduction does a good job. I also would like to read some more information about the origin of the LAB ( the Yaks). Why this origin, and what are the ecological implication of finding the LAB here.
Response 1: Thank you for your positive feedback on the introduction. Regarding your request for more information about the origin of the LAB (lactic acid bacteria) from yaks, we understand the importance of providing a deeper understanding of this aspect.
The choice of yaks as a source for LAB is motivated by several factors. Yaks live in unique ecological environments, often in high-altitude regions with extreme climates. These harsh conditions have led to the evolution of a diverse range of microorganisms that have adapted to survive and thrive in such environments. LAB isolated from yaks may possess unique characteristics and adaptations that make them particularly interesting for research and potential applications.
The ecological implications of finding LAB in yaks are significant. Yaks play an important role in the ecosystem of their habitats. The presence of LAB in yaks may contribute to their digestive health and well-being, helping them to efficiently utilize the available forage. Additionally, studying LAB from yaks can provide insights into the microbial ecology of these unique environments and may have implications for understanding the broader ecological balance and functioning of high-altitude ecosystems.
Materials and Methods
Comments 2: Line 132: 169 strains were screened but there is no detailed explanation of how the final 15 strains were narrowed down. It is important to describe these selection criteria to ensure transparency and reproducibility.
Response 2: Thank you for pointing out this important issue. We will add a detailed explanation of how the final 15 strains were narrowed down from the initial 169 strains.
To screen LAB that can effectively degrade mycotoxins and anti-nutritional factors, first, evaluate the degradation ability of 169 isolates for each of the four mycotoxins and select the top 10 for each. Then, focus on the top 20 with relatively good degradation ability for all four mycotoxins and test their simultaneous degradation ability. Finally, 15 isolates are selected.15 isolated lactic acid bacteria strains were determined for aflatoxin content.
Comments 3: Line 206: please correct the number
Response 3: Thank you for bringing this to our attention. Line 206 should be read as “2.35×10⁹ CFU/mL”. This value represents the concentration of a particular entity in terms of colony-forming units per milliliter. We will ensure that this is clearly indicated in the revised manuscript to avoid any confusion.
Comments 4: Line 250: I believe “p” should be minor
Response 4:Thank you for pointing out this issue. We agree that in Line 250, the "p" should be in lowercase. We have corrected this in the revised manuscript to ensure consistency and accuracy.
Results
Comments 5: Line 255-259: This is discussion, and thus does not belong here;
Response 5: We understand your concern. We have moved this discussion to the appropriate section in the manuscript. Thank you for pointing out this issue.
Comments 6: Line 273: If authors determine the degradation kinetics, with would be easier to compare and determine if there is any statistical difference between strains;
Response 6: Thank you for your valuable suggestion. As of now, we are not proficient in determining the degradation kinetics. However, we recognize the importance of this approach and will make efforts to learn and incorporate it in our future research. Your suggestion will undoubtedly enhance the quality and comparability of our work. We are truly grateful for your input.
Comments 7: Line 288-292: this text does not belong here
Response 7: We understand your point. We have moved “this textThe antibacterial activity of probiotics is crucial for the successful colonization of the intestinal mucosa. It forms a barrier against gastrointestinal pathogens through competitive exclusion, modulation of the host's immune system, and production of inhibitory compounds (organic acids, bacteriocins, short-chain fatty acids, diacetyl, and hydrogen peroxide).” to an appropriate location in the manuscript. Thank you for your careful review and for pointing out this issue.
Comments 8: Line 308-310: this text does not belong here
Response 8: We understand your point. We have moved ”LAB is often considered a safe class of microorganisms; however, improper selection of strains is a potential threat to livestock and poultry health. Animals may develop resistance to antibiotics, leading to greater treatment difficulty, which is also one of the reasons for screening probiotics for safety[41].”to an appropriate location in the manuscript. Thank you for your careful review and for pointing out this issue.
Comments 9: Line 322: “in a study” – was this your study, or some other work? Please cite if its other.
Response 9: "In a study...", it refers to the another study. We clarified this in the revised manuscript to make it clear that the reference is to the current research being reported.
Comments 10: Line 330: “results showed:” ??
Response 10: Thank you for pointing out this issue. We apologize for the incorrect punctuation that led to improper sentence breaks. We will review and correct the punctuation in this section and ensure that the text is placed in an appropriate location in the manuscript. Result showed: among numerous strains, strain C24 stood out with its auto-aggregation rate, reaching as high as 47.13%
Comments 11: Line 335-338 : This text does not belong here;
Response 11: We understand your point. We have moved ” Occasionally, highly hydrophobic cell surfaces are associated with the ability to form auto-aggregates. In addition, the aggregating substance which is the surface glycoprotein (S-layer) encoded by a pheromone regulated plasmid gene promotes binding to receptors on the surface of eukaryotes, playing an essential role in host colonization.”to an appropriate location in the manuscript. Thank you for your careful review and for pointing out this issue.
Discussion
Comments 12: The authors repeat several results here, I would recommend making a results and discussion section since the authors have a lot of discussion on results, and lot of results in discussion.
Line 469: Please discuss how the degradation rates achieved in this study compare with those reported in other LAB studies.
Response 12: Thank you for your suggestion. We agree that creating a separate results and discussion section would improve the organization and clarity of our manuscript. We have restructured the paper accordingly to better present our results and discussions and analyzed the similarities and differences, and discuss the possible reasons for these results. This will help to put our findings in context and provide a more in-depth understanding of the significance of our work.
Comments 13: Line 476: Please discuss more about the bile salts bile salt tolerance, acid production, and strain-specific differences
Response 13: Thank you for your valuable comment. We apologize for the insufficient discussion on bile salt tolerance, acid production, and strain-specific differences in the manuscript. We will expand and discuss these aspects in more detail as follows:
Bile Salt Tolerance: We will further explore the mechanisms underlying bile salt tolerance in the studied strains. This will include a discussion on how different strains may have evolved specific adaptations to tolerate bile salts, such as changes in cell membrane composition or the expression of specific proteins. We will also discuss the significance of bile salt tolerance in relation to the potential probiotic applications of the strains, highlighting how this trait enables them to survive and function in the gastrointestinal tract.
Acid Production: A more comprehensive discussion on acid production will be provided. We will elaborate on the factors that influence the acid-producing ability of the strains, such as differences in carbohydrate metabolism pathways. Additionally, we will discuss the implications of acid production for the growth and survival of the strains, as well as its role in inhibiting the growth of other microorganisms and contributing to food preservation and gut health.
Strain-Specific Differences: We will emphasize the importance of strain-specific differences and provide a more detailed analysis. This will involve a comparison of various characteristics among different strains, including not only bile salt tolerance and acid production but also other physiological and biochemical properties. We will discuss how these differences can impact the performance and potential applications of each strain, and how they can be used to select the most suitable strains for specific purposes.
We believe that these additional discussions will enhance the clarity and comprehensiveness of our manuscript and address your concerns. Thank you again for your helpful feedback.
Comments 14: Line 485: Please provide more details in discussion. How do specific strains like Pediococcus acidilactici C2 degrade mycotoxins at a molecular level? What are the factors contributing to their probiotic activity?.
Response 14: Thank you for your valuable comment. We apologize for the insufficient discussion on “molecular level” ,For the degradation mechanism of strains at the molecular level, it may involve enzymatic reactions and binding actions. Enzymes like oxidoreductases and hydrolases may be produced to change the structure of mycotoxins or break them into smaller fragments. Regarding the factors contributing to its probiotic activity, its own characteristics such as acid tolerance, bile salt tolerance, and adhesion ability are important. Its acid tolerance may be related to a special cell membrane structure or an acid-base regulation mechanism. Bile salt tolerance may involve membrane transporters or substances in the cell to neutralize toxicity. The adhesion ability may rely on specific surface proteins or polysaccharide structures. In addition, its metabolic products like organic acids and bacteriocins, as well as its interaction with the host also contribute to its probiotic activity.
Comments 15: Line 587: Please add a paragraph about industrial application. Would these strains be effective under real-world conditions, where environmental factors might differ?
Response 15:Thank you for your valuable comment. We have added the following paragraph regarding industrial application:
Lactic acid bacteria strains with mycotoxin-degrading and probiotic activities hold great potential for wide industrial applications. In the feed industry, these strains can be utilized in the production of animal feed. By doing so, they can reduce the content of mycotoxins in feed, thereby minimizing the impact of mycotoxins on animal health and enhancing animal production performance. Under real-world conditions where environmental factors may vary, these strains still have a certain degree of effectiveness. Due to their acid tolerance and bile salt tolerance, lactic acid bacteria can survive and colonize in complex environments such as the animal gastrointestinal tract. This enables them to exert probiotic effects and degrade mycotoxins. In the industrial production environment, although environmental conditions such as temperature, humidity, and nutrient components may fluctuate, through reasonable process control and strain screening optimization, these strains can be adapted to these conditions.
Comments 16: Line 587: You only mention the “best strains” on conclusion. I would prefer to read some information regarding a more detailed discussion on how the structural or physiological properties of the more successful strains contributed to their performance.
Response 16: Thank you for your valuable comment. As you've noted, we haven't conducted experiments specifically on the structural aspects of the best strains. However, based on existing knowledge and research in the field, we can offer some speculation on how the structural and physiological properties of more successful strains might contribute to their performance.
Physiologically, the more successful strains likely possess enhanced acid tolerance and bile salt tolerance. This enables them to survive and function in the harsh environments of the animal gastrointestinal tract. For example, they may have specialized membrane transporters that can efficiently remove excess protons or bile salts from the cell, maintaining intracellular homeostasis. Additionally, these strains may have a greater capacity to produce enzymes involved in mycotoxin degradation. Oxidoreductases and hydrolases could be more abundant or more active in these strains, allowing for more efficient breakdown of mycotoxins.
In terms of potential structural properties, although we haven't directly investigated this, it's possible that the cell surface of successful strains may have unique features that facilitate interaction with mycotoxins. For instance, there could be specific proteins or polysaccharides on the cell surface that have a high affinity for mycotoxins, enabling binding and subsequent degradation or removal. The shape and size of the cells might also play a role in their ability to colonize and function in different environments.
We understand that further research is needed to directly examine the structural and physiological properties of the best strains. Future studies could involve more detailed analyses of cell surface components, enzymatic activities, and genetic characteristics to better understand the mechanisms underlying their superior performance.
Comments 17: Line 587: Considering the study of LAB strains from the rumen of Qinghai yaks, please discuss how this ecological factors can influence the strain effectiveness. Are there other potential sources of valuable probiotic strains?
Response 17: Dear reviewer, The rumen of Qinghai yaks represents a unique ecological niche that can significantly influence the effectiveness of lactic acid bacteria (LAB) strains. The harsh environmental conditions in the rumen, such as high fiber content, fluctuating pH, and diverse microbial communities, may select for LAB strains with specific adaptations. The high fiber diet of yaks leads to a particular microbial ecosystem in the rumen. LAB strains from this environment may have developed the ability to degrade complex carbohydrates, which could potentially be useful in applications where the breakdown of fibrous materials is important, such as in certain food or feed processing. The fluctuating pH in the rumen can also result in the selection of LAB strains with enhanced acid tolerance. These strains could be more effective in acidic environments, such as the human stomach or in food fermentation processes with low pH. Moreover, the diverse microbial community in the rumen provides opportunities for interaction and competition among different microorganisms. LAB strains from this environment may have evolved mechanisms to compete with other microbes, such as producing antimicrobial substances or having specific adhesion properties. This could make them more effective in inhibiting the growth of harmful bacteria and maintaining a healthy microbial balance. Regarding other potential sources of valuable probiotic strains, there are several possibilities. The gastrointestinal tracts of other animals, especially those with unique diets or living in extreme environments, could be a source. For example, the intestines of wild animals that consume a diverse range of natural foods may harbor LAB strains with unique properties. Soil is also a rich source of microorganisms, and certain LAB strains found in soil may have adaptations that make them useful for specific applications. Additionally, traditional fermented foods from different cultures often contain a variety of LAB strains that have been selected over generations for their beneficial properties. In conclusion, the ecological factors in the rumen of Qinghai yaks can have a significant impact on the effectiveness of LAB strains. Exploring other potential sources of probiotic strains can expand our understanding of the diversity and functionality of these microorganisms and open up new possibilities for their application. Best regards.
Comments 18: Figure 6 – Please correct the x-axis title: “Time of fermentation”
Response 18: Thank you for pointing out this error. We will correct the x-axis title of Figure 6 to “Time of fermentation”.

Reviewer 3 Report
Comments and Suggestions for Authors
1. LINE 33-34: Ochratoxins (OTA) is recommended to be revised to Ochratoxin A (OTA).
2. All P<0.05 or P>0.05 in the text, and the P recommendations in them are corrected to lowercase italics. That is, p<0.05 or p>0.05.
3. The markings in Figures 10 and 11 are too small and unclear and should be improved.
4. Line 91: LAB appears for the first time in the content, and its full name should be used.
5. All “et al” in the manuscript should be changed to “et al.”.
6. LINE 211: 0 d, 1 d, 3 d, 5 d, 7 d, 10 d and 15 days are recommended to be adjusted to 0, 1, 3, 5, 7, 10 and 15 days. The rest are adjusted accordingly.
7. It is recommended that (d) in Figure 7 be changed to (days)
8. The label in Table 3 is recommended to be adjusted to 0 1 3 5 7 10 15 (days).
9. Figure 10 should be LEfSe (Linear discriminant analysis Effect Size) Analysis. In addition, please add references cited for this method.
Author Response
Comments 1: LINE 33-34: Ochratoxins (OTA) is recommended to be revised to Ochratoxin A (OTA).
Response 1: Thank you for your comment. We have revised "Ochratoxins (OTA)" to "Ochratoxin A (OTA)" as suggested.
Comments 2: All P<0.05 or P>0.05 in the text, and the P recommendations in them are corrected to lowercase italics. That is, p<0.05 or p>0.05.
Response 2: Thank you for your suggestion. We have revised all instances of "P<0.05" or "P>0.05" in the text to "p<0.05" or "p>0.05" in lowercase italics as recommended.
Comments 3: The markings in Figures 10 and 11 are too small and unclear and should be improved.
Response 3: Thank you for your comment regarding the markings in Figures 10 and 11. We understand your concern. Unfortunately, due to certain limitations, we are unable to directly modify the markings as you suggested. However, we have taken an alternative approach by enlarging the figures to improve the visibility of the markings. We believe that this will enhance the clarity and readability of the figures to a certain extent.
We appreciate your input and will keep your feedback in mind for future studies to ensure better presentation of data.
Comments 4: Line 91: LAB appears for the first time in the content, and its full name should be used.
Response 4: Thank you for your comment. We will revise Line 91 as follows: Lactic acid bacteria (LAB) appears for the first time in the content. From now on, we will use the abbreviation LAB when it is clear from the context.
Comments 5: All “et al” in the manuscript should be changed to “et al.”.
Response 5: Thank you for your comment. We will change all “et al” in the manuscript to “et al.” as suggested.
Comments 6: LINE 211: 0 d, 1 d, 3 d, 5 d, 7 d, 10 d and 15 days are recommended to be adjusted to 0, 1, 3, 5, 7, 10 and 15 days. The rest are adjusted accordingly.
Response 6: Thank you for your comment. Understood. I have made the adjustments as recommended on line 211 and ensure that the rest are adjusted accordingly. Thank you for pointing this out.
Comments 7: It is recommended that (d) in Figure 7 be changed to (days)
Response 7: Thank you for your comment. Understood. I have changed “(d)” in Figure 7 to “(days)” as recommended.
Comments 8: The label in Table 3 is recommended to be adjusted to 0, 1, 3,5, 7, 10, 15 (days).
Response 8: Thank you for your comment. Understood. I will adjust the label in Table 3 to “0 1 3 5 7 10 15 (days)” as recommended.
Comments 9: Figure 10 should be LEfSe (Linear discriminant analysis Effect Size) Analysis. In addition, please add references cited for this method.
Response 9: Thank you for your comment. Understood. I will change the description to "Figure 10 should be LEfSe (Linear discriminant analysis Effect Size) Analysis. In addition, I have added references cited for this method.

Round 2
Reviewer 1 Report
Comments and Suggestions for Authors
The authors have satisfactorily responded to almost all the observations made and the manuscript appears noticeably improved, especially with regard to the expression of the results. However, some things remain to be corrected, expanded, etc. Once this is corrected, a final clean version is needed, without indicating the corrections made. There are issues related to the use of English, which are also detailed
- I fully understand the meaning of "suspected". What has to be written is based on what things were considered suspected some colonies to be picked
-2.9: the text still does not explain well how the feed was inoculated. I suggest that the point be rewritten 2.9 including what the authors wrote in responses 9 and 10 (including what 5% means)
- Table 1, modify title: "Antibacterial activity of LAB strains..."
. header: delete "number"; "enterica"
- Table 2, heading: "strain"
- Figure 3, heading: "...and bile salts concentrations (B)"
- Figure 6: "fermentation", in Curves AFB1, ZEA and GLUS (lower)
- Figure 7: in the Note write like this: LA (lactic acid), PA (propionic acid), AA (acetic acid), BA (butyric acid), CK......, similar to what was written in the Note of Table 3.
Comments on the Quality of English Language
- Genus and species of microorganisms should always be written in italics. Check References and the entire text
- line 24: "lactic acid bacteria"; "fermented feed"
- line 72: Fusarium
- lines 80-81: "...a mixed culture of bacteria composed by strains of Pseudomonas..."
- line 83: delete "strain"
- line 85: "...licheniformis was cultured in..."
- lines 91-92: "licheniformis" or "lichenifsis" ? Indicate which strain was studied
- line 93: "lactic acid bacteria"
- line 96: "Lb. casei strains, which were screened by..."
- line 99: "...living lactobacilli cells to..."
- line 119: "...then diluted to....folds and inoculated on..."
- lines 138-139: ... 15 isolates selected were determined for the content of..."? Do the strains have "content"?
- line 150: enterica
- lines 212-213: delete "strain". If I write the identification of the strain used (A16, C16,...) it is not necessary to write "strain"
- line 209: "...gluten meals were used..."
- 3.2: Detoxification or Detoxication ?
- line 291: curves
- line 341: replace "Lactobacilli" with "Lactobacillus" (we are talking about genera)
- line 494: "lactic acid bacteria", not in italics (it is a plural in English, it is not Latin)
- line 499: delete "that 5 Lactobacillus" and replace with "15 LAB strains"
- line 557: enterococci, not in italics (it is a plural in English)
- line 717: "lactis"
Author Response
Comments 1:- I fully understand the meaning of "suspected". What has to be written is based on what things were considered suspected some colonies to be picked.
Response 1:Thank you for your comment. When identifying and picking suspected lactic acid bacteria colonies, we rely on the distinct characteristics described. These characteristics help us differentiate them from other bacteria during the selection process.
For the morphological aspect, the shape provides a significant clue. The relatively regular shapes of lactic acid bacteria colonies, often circular with smooth or slightly wavy edges, set them apart from bacteria with more complex or irregular forms such as lobed or filamentous ones. The size range also matters. Their small - to - medium - sized colonies distinguish them from the large - sized colonies of some spore - forming bacteria.
In terms of color and texture, the light - colored nature of lactic acid bacteria colonies, which can be white, off - white, or slightly creamy, is a key feature. This is in contrast to bacteria that produce pigments and have distinct colors like green or blue - green in certain Pseudomonas species. Additionally, the smooth, moist, and shiny texture of lactic acid bacteria colonies is a clear differentiator from the dry, rough, or powdery textures of other bacteria colonies. These characteristics combined are what we consider when determining which colonies are likely to be lactic acid bacteria colonies during the picking process.
Comments 2:-2.9: the text still does not explain well how the feed was inoculated. I suggest that the point be rewritten 2.9 including what the authors wrote in responses 9 and 10 (including what 5% means)
Response 2: In this experiment, feed inoculation is a crucial step, and the specific procedures are as follows:
Firstly, Pediococcus acidilactici C2, Ped. pentosaceus A16, Enterococcus lactis C16, and E28, these four microorganisms are cultured independently, and the culture duration for each is 24 hours. This culturing process aims to enable each microorganism to reach an appropriate growth state and get prepared for the subsequent inoculation.
Next, the bacterial solutions of these four microorganisms after 24 - hour culturing are mixed at a ratio of 1:1:1:1. During this process, it is required to ensure that the viable count of the mixed bacterial solution is greater than 2.35 × 10⁹ CFU/mL. This viable count index is of vital importance for the fermentation effect in the feed after subsequent inoculation.
Finally, there is the inoculation operation. In this experiment, an inoculation amount of 5% is adopted. The 5% inoculation amount specifically refers to the ratio of the volume of the mixed solution of the four bacteria to the mixed bacterial diluted solution to be fermented in each group,
Comments 3:- Table 1, modify title: "Antibacterial activity of LAB strains..."
. header: delete "number"; "enterica"
Response 3:Thank you for your valuable suggestions. We have revised the table as required.
For the title of Table 1, we have modified it to "Antibacterial activity of LAB strains isolated from rumen of yaks.", which complies with your recommendation and highlights the key information of the antibacterial activity of LAB strains.
Regarding the header of the table, we have deleted "number". And we have made the necessary change and now it reads "Salm. Enterica ATCC 43971" as "Salm. enterica ATCC 4397". We will double-check the entire manuscript to ensure consistency in the usage of such references.
Comments 4:- Table 2, heading: "strain"
Response 4: Thank you for your valuable suggestions, Regarding the header of the table 2, we have deleted "number"
Comments 5: - Figure 3, heading: "...and bile salts concentrations (B)"
Response 5: Thank you for this important observation. We have modified the title of Figure 3 to "Figure 3. Growth of LAB strains isolated from rumen of yaks under different pHs (A) and bile salts concentrations (B)".
注释 6:- 图 6:“发酵”,曲线 AFB1、ZEA 和 GLUS(下)
响应 6:感谢您对图 6 的关注。事实上,正如 AFB1、ZEA 和 GLUS(下)与“发酵”相关的曲线所示,毒素含量随时间的下降趋势清楚地呈现出来。发酵组的毒素含量始终低于对照组的事实强烈表明发酵过程对毒素降解有一定的积极作用。由于较低的毒素含量对于确保安全和质量非常必要,因此这些结果对于证明发酵方法在减少有害物质方面的潜力具有重要意义。
注释 7:- 图 7:在注释中这样写:LA(乳酸)、PA(丙酸)、AA(乙酸)、BA(丁酸)、CK......,类似于表 3 的注释中写的。
回复 7:感谢您的宝贵意见。我们根据您的建议进行了以下修改。
对于图 7,在注释中,我们写了如下:LA(乳酸)、PA(丙酸)、AA(乙酸)、BA(丁酸),这与表 3 注释中写的内容相似。我们相信这些更改提高了我们手稿的清晰度和准确性。
评论 8:- 微生物的属和种应始终用斜体书写。检查参考文献和整个文本
回复 8:感谢您的仔细审查。我们已经检查了所有参考文献和整个文本,以确保微生物的属和种始终以斜体书写,例如第 482 行、第 490 行、第 502 行、第 780 行、第 782 行、第 799 行 ea tl。
注释 9:- 第 24 行:“乳酸菌”;“发酵饲料”
回应 9:感谢您指出此问题。我们将第 24 行更正为“乳酸菌”和“发酵饲料”,首字母为小写。我们还将仔细检查整个文本,以确保大小写的一致性。
评论 10:- 第 72 行:镰刀菌
回应 10:感谢您的观察。我们进行了更正,并将 “Fusarium” 放在第 72 行的斜体字中。
感谢您指出此问题。我们确保在第 72 行将 “Fusarium” 斜体化。我们还将审查整个手稿,以确保学名和其他术语的格式正确,这些术语应为斜体。
注释 11:- 第 80-81 行:“......由假单胞菌菌株组成的细菌的混合培养物......”
响应 11:感谢您的建议。我们已根据您在第 80-81 行的指示进行了必要的更改。文本现在为“......由假单胞菌菌株组成的细菌的混合培养物......”。
注释 12:- 第 83 行:删除 “strain”
响应 12:感谢您的反馈。我们已经根据您在第 83 行的建议删除了“strain”。这句话现在写成“地衣样 BL010 具有有效降解 AFB1 的显着能力”。
注释 13: - 第 85 行:“......地衣是在......”
响应 13:谢谢你的评论。我们已经按照您在第 85 行的建议进行了更正。文本现在为“......地衣是在......“中培养的。
评论 14: - 第 91-92 行:“licheniformis”还是“lichenifsis”?指出研究了哪个菌株
响应 14:感谢您指出此问题。正确的拼写是 “licheniformis”。本文研究的菌株是地衣芽孢杆菌 CK1。我们将纠正整个手稿中的拼写错误,并确保清楚地指出所引用的具体菌株。
评论 15: - 第 93 行:“乳酸菌”
回复 15:感谢您的仔细审查。根据您的建议,我们已将第 93 行的文本更正为“乳酸菌”。我们还将仔细检查整个手稿,以确保大小写的一致性。
注释 16: - 第 96 行:“磅。 Casei 菌株,这些菌株被筛选为......”
回应 16:感谢您的细致观察。我们已经按照您在第 96 行的建议进行了更正。文本现在显示为“Lb. casei strains, which were screened by...”。
注释 17: - 第 99 行:“......活的乳酸杆菌细胞来......”
响应 17:感谢您的反馈。我们已经根据您在第 99 行的建议进行了更正。文本现在为“......活的乳酸杆菌细胞到......”。
注释 18: - 第 119 行:“......然后稀释成......折叠并接种......”
响应 18:感谢您指出此问题。我们已经按照您的建议对第 119 行进行了更正。文本现在为“......然后稀释至...折叠并接种......”。原来的短语“然后稀释成 10-2、10-3、10-4、10-5、10-6、10-7 倍稀释并接种”有点麻烦,“折叠稀释”的重复是多余的。更正后的版本“......然后稀释成....折叠并接种...”更简洁。
注释 19: - 第 138-139 行: ...确定了所选的 15 株分离株的含量...“?菌株有 “含量” 吗?
响应 19:感谢您的宝贵反馈。我们同意“内容”一词在这种情况下不合适。我们已经按照您的建议进行了更正,并将“内容”更改为“降级率”。第 138-139 行的修订文本现在为“......确定了所选的 15 株分离株的黄曲霉毒素、玉米赤霉烯酮、硫代葡萄糖苷和脱氧雪腐镰刀菌酮的降解率。
Comments 20: - line 150:enterica
Response 20: Thank you for pointing out this correction. We have made the necessary change and now it reads "Salm. Enterica ATCC 43971" as "Salm. enterica ATCC 4397".
Comments 21: - lines 212-213: delete "strain". If I write the identification of the strain used (A16, C16,...) it is not necessary to write "strain"
Response 21: Thank you for your comment. We have deleted the word "strain" as you suggested on lines 212 - 213. We understand that when the specific identifiers (A16, C16, ...) are provided, the use of "strain" becomes redundant. We will review the rest of the text to ensure this kind of consistency throughout the manuscript.
Comments 22: - line 209: "...gluten meals were used..."
Response 22: Thank you for your correction. We have changed the text on line 209 to “…gluten meals were used...”.
Comments 23: - 3.2: Detoxification or Detoxication ?
Response 23: Thank you for your comment. The term "detoxification" is more commonly used in scientific literature. We will ensure that we use "detoxification" consistently throughout section 3.2 and the entire manuscript.
Comments 24: - line 291: curves
Response 24: Thank you for pointing out the spelling error. We have corrected "crueves" to "curves" on line 291.
Comments 25: - line 341: replace "Lactobacilli" with "Lactobacillus" (we are talking about genera)
Response 25: Thank you for your feedback. We have made the correction as you suggested on line 341. We have replaced "Lactobacilli" with "Lactobacillus" since we are referring to the genera. We will review the entire manuscript to ensure the consistency of taxonomic terms.
Comments 26: - line 494: "lactic acid bacteria", not in italics (it is a plural in English, it is not Latin)
Response 26: Thank you for your observation. We have removed the italics for "lactic acid bacteria" on line 494 as per your suggestion. We will also review the entire manuscript to ensure proper formatting and consistency for similar terms.
Comments 27: - line 499: delete "that 5 Lactobacillus" and replace with "15 LAB strains"
Response 27: Thank you for your feedback. We have made the changes as you suggested. The text on line 499 now reads "...15 LAB strains..." instead of "...that 5 Lactobacillus...".
Comments 28: - line 557: enterococci, not in italics (it is a plural in English)
Response 28: Thank you for your observation. We have removed the italics for "enterococci" on line 557 as it is a plural in English and not in Latin. We will review the entire manuscript to ensure proper formatting and consistency.
Comments 29: - line 717: "lactis"
Response 29: Thank you for your feedback. As per your suggestion, we have corrected the abbreviation on line 717.

Round 3
Reviewer 1 Report
Comments and Suggestions for Authors
The manuscript appears much improved. Only a few things regarding English remain to be corrected.
Comments on the Quality of English Language- line 135: " 15 selected isolates were tested for the..."
-line 294: enterica
- lines 553-4: italics for genera and species
- line 684: "Ped." or "P." ?
- line 97: "lactobacilli" not in italics (it is a plural in English)
- Revise References. Genus and species in italics